

**Spatial-temporal changes in flow hydraulic characteristics and soil loss**
**during gully headcut erosion**
Mingming Guo[a], Zhuoxin Chen[b], Wenlong Wang[b,c*], Tianchao Wang[d], Qianhua Shi[b], Hongliang
Kang[b], Man Zhao[b], Lanqian Feng[c]
a Key laboratory of Mollisols Agroecology, Northeast Institute of Geography and Agroecology,
Chinese Academy of Sciences, Harbin 150081, Heilongjiang, China
b State Key Laboratory of Soil Erosion and Dryland Farming on the Loess Plateau, Institute of Water
and Soil Conservation, Northwest A&F University, Yangling, Shaanxi 712100, China
c Institute of Soil and Water Conservation, Chinese Academy of Sciences and Ministry of Water
Resources, Yangling, Shaanxi 712100, China
d Ulanyab Grassland Station, Jining, Inner Mongolia 012000, China
**\*Corresponding author:** Wenlong Wang
E-mail addresses: nwafu_wwl@163.com; wlwang@nwsuaf.edu.cn





## Abstract

The temporal-spatial changes in flow hydraulics and energy consumption and their associated soil erosion remain unclear during gully headcut retreat. A simulated scouring experiment was conducted on five headcut plots consisting of upstream area (UA), gully headwall (GH) and gully bed (GB) to elucidate the temporal-spatial changes in flow hydraulic, energy consumption, and soil loss during headcut erosion. The flow velocity at the brink of headcut increased as a power function of time, whereas the jet velocity entry to plunge pool and jet shear stress logarithmically or linearly decreased over time. The jet properties significantly affected by upstream flow discharge. The Reynold number, runoff shear stress, and stream power of UA and GB increased as logarithmic or power functions of time, but the Froude number decreased logarithmically over time. The flow of UA and GB was supercritical and subcritical, respectively, and transformed to turbulent with inflow discharge increased. The Reynold number, shear stress and stream power decreased by 56.0%, 63.8% and 55.9%, respectively, but the Froude number increased by 7.9% when flow dropped from UA to GB. The accumulated runoff energy consumption of UA, GH and GB positions linearly increased with time, and their proportions of energy consumption are 18.3%, 77.7% and 4.0%, respectively. The soil loss rate of the "UA-GH-GB" system initially rose and then gradually declined and levelled off. The soil loss of UA and GH decreased logarithmically over time, whereas the GB was mainly characterized by sediment deposition. The proportion of soil loss at UA and GH are 11.5% and 88.5%, respectively, of which the proportion of deposited sediment on GB reached 3.8%. The change in soil loss of UA, GH and GB was significantly affected by flow hydraulic and jet properties. The critical energy consumption initiating soil erosion of UA, GH, and GB are 1.62 J s$^{-1}$, 5.79 J s$^{-1}$ and 1.64 J s$^{-1}$, respectively. These results are helpful to reveal the mechanism of gully headcut erosion and built headcut migration model.

**Keywords:** Gully erosion; Hydraulic property; Headcut retreat; Bank collapse; Loess Plateau



## 1 Introduction

Gully erosion is a typical soil erosion process whereby concentrated runoff from an upstream drainage area recurs in a channel and erodes soil from the area through which runoff passed to considerable depth (Poesen et al., 2003; Zhu, 2012). Gully erosion is recognized as the main sediment source in some hilly and gully-dominated watersheds (Poesen et al., 2003; Valentin et al., 2005; Dotterweich et al., 2012). Poesen et al. (2003) reported that soil loss amount caused by gully erosion accounts for 10% - 94% of total soil loss amount based on the collected data from published articles. Moreover, gully erosion can severely damage to infrastructure, enhance the terrain fragmentation, and cause ecosystem instability, land degradation and food safety (Poesen et al., 2003; de Vente & Poesen, 2005; Li et al., 2015; Vanmaercke et al, 2016; Hosseinalizadeh et al., 2019).

As one of the gully erosion processes, the gully headcut retreat often significantly influences and determines gully erosion (Oostwoud-Wijdenes et al., 2000; Vandekerckhove et al., 2003; Guo et al., 2019). A headcut is defined as a vertical or near-vertical drop or discontinuity on the bed of a gully occurring where flow is concentrated at a knickpoint (Hanson et al., 2001; Bennett et al., 2000). Many studies have demonstrated that the gully erosion is the result of the combined actions of plunge pool erosion by jet flow, upstream runoff incision, headwall erosion by on-wall flow, mass failure (gully head and wall collapse), (Vanmaercke et al., 2016; Addisie et al., 2017; Guo et al., 2019). Once a headcut is formed in upstream area, the gully will develop rapidly and not stop forward until a critical topographic condition is formed ($S \leq a \cdot A^b$, where $S$ and $A$ is the slope gradient and drainage area upstream gully headcut, respectively) (Kirkby et al., 2003). Moreover, in fact, the erosion processes of different landform units (upstream area, UA; gully head, GH; gully bed, GB) are completely different during gully headcut erosion (Zhang et al., 2018; Guo et al., 2019; Shi et al., 2020a). The combination and interaction of erosion processes of the three landform units determined gully headcut erosion process (Vanmaercke et al., 2016). Therefore, clarifying the soil erosion process and characteristics of the three landform units is critical to systematically and clearly reveal the mechanism of gully headcut erosion.

Previous studies suggested that gully heacut erosion is affected by various factors including topography, land use change, vegetation, soil properties, and climate (Vanwalleghem et al., 2003;





Ionita, 2006; Rodzik et al., 2009; Rieke-Zapp and Nichols, 2011; Torri and Poesen, 2014; Ionita et al.,
2015; Vannoppen et al., 2015; Guo et al., 2019, 2020a). In terms of topography, most of studies
focused on the threshold relationship ($S \leq a \cdot A^b$) to initiate gully erosion (e.g. Torri and Poesen, 2014).
Several experimental studies demonstrated that the upstream slope gradient and headcut height have
significant effects on headcut erosion (e.g. Bennett, 1999; Zhang et al., 2018). Land use change is
recognized as having the strongest effect on processes related to gully erosion among influencing
factors (Poesen et al., 2003; Chaplot et al., 2005; Descroix et al., 2008), and also significantly affects
the activation of gully headcut erosion (e.g. Torri and Poesen, 2014). In this aspect, the vegetation
coverage is a parameter that is often used to clarify its effect on gully erosion (e.g. De Baets et al.,
2007; Martínez-Casasnovas et al., 2009), however, in fact, the vegetation effect mainly depended on
the root characteristics and its distribution at gully head (e.g. Vannoppen et al., 2015; Guo et al.,
2019). Nevertheless, at present, the most of studies on gully erosion focus on the changes in gully
morphology between different periods at a watershed or regional scale (Vanmaercke et al., 2016),
which is why the previous studies fail to address the effects of root systems on gully headcut retreat.
Guo et al. (2019) concluded that the grass (*Agropyron cristatum*) could reduce soil loss and headcut
retreat distance by 45.6–68.5%, 66.9–85.4%, respectively, and the roots of 0–0.5 mm in diameter
showed the    greatest controlling influence on headcut erosion. In terms of soil properties, lots of
studies have proved the significant effect of soil properties on gully headcut erosion (e.g. Nazari
Samani et al., 2010), which was mainly related to the change in soil erodibility induced by soil
properties including soil texture, soil vertical joints, soluble mineral content, soil lithology, and
physicochemical properties (Sanchis et al., 2008; Vanmaercke et al., 2016; Guo et al., 2020a).
Rainfall, the main climate factor, is closely related to runoff generation and thus be expected to affect
headcut erosion. Many studies have reported that the initiation of gully headcut is correlated with
rainfall characteristics (e.g. summation of rainfall from 24-hour rains equal to or greater than 0.5
inches) (Beer and Johnson, 1963; Vandekerckhove et al., 2003; Rieke-Zapp and Nichols, 2011).
However, the great difference in the threshold value relating to rainfall factors was found among
different areas of the world due to great difference in erosion environment. For example, in the
northeast of China, the gully erosion is the result of soil thawing, rainfall runoff and snowmelt runoff

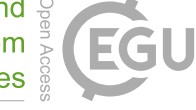

(Li et al., 2016b; Xu et al., 2019). Furthermore, at present, the most of studies on gully erosion were
conducted to quantify the change in gully erosion (retreat rate, area and volume) at different spatial
and temporal scales by using remote sensing interpretation, real-time monitoring and meta-analysis
based on literature data (e.g. Vanmaercke et al., 2016). However, the influencing mechanism of these
factors on gully headcut erosion is still unclear and need to be revealed in future studies.
Evidently, the concentrated flow upstream gully head, mainly depended on the upstream area
and rainfall, is the main and original drive force triggering headcut erosion. The runoff firstly eroded
the upstream area and then was parted into two types of flow (on-wall flow and jet flow) at the
brinkpoint of gully headcut. Consequently, the on-wall flow persistently eroded the headwall soil,
and the jet flow violently impacted gully bed soil and formed a plunge pool (Su et al., 2015; Guo et
al., 2019). Subsequently, the two types of flow merged again and eroded gully bed together (Zhang et
al., 2018; Shi et al., 2020a). The runoff hydraulic or jet flow properties at different landform units
(upstream area, gully head and gully bed) are significantly different, which is an important reason for
the difference in erosion process among different landform units. However, the temporal-spatial
change in runoff and jet properties during headcut erosion is still unclear and thus needs to be
clarified. Furthermore, at present, some experimental studies on headcut erosion of rill, ephemeral
gully, gully and bank gully were conducted to investigate the runoff properties, energy consumption,
sediment transport process, morphology evolution and empirical model (Bennett and Casalí, 2001;
Wells et al., 2009a, 2009b; Su et al., 2014; Xu et al., 2017a; Guo et al., 2019; Shi et al., 2020a).
However, relatively few knowledges were obtained to systemically reveal the hydrodynamic
mechanism of gully headcut erosion. Therefore, elucidating the temporal-spatial changes in runoff
hydraulic and soil loss and hydrodynamic mechanism of UA, GH and GB is of great importance to
systematically reveal the hydrodynamics mechanism of gully headcut erosion.
Given the above-mentioned issues, a series of simulated gully headcut erosion experiments
subjected to inflow scouring are conducted to (1) investigate the temporal-spatial change in hydraulic
properties and soil loss during headcut erosion, (2) quantify the energy consumption and soil loss
distribution of UA, GH and GB, and (3) reveal the erosion hydrodynamic mechanism of UA, GH
and GB.



## 2 Materials and Methods

### 2.1 Study area

This experiment was carried out at the Xifeng Soil and Water Conservation Experimental Station that is located in the Nanxiaohegou watershed, Qingyang City, Gansu Province, China (Fig. 1). The study area belongs to a semi-arid continental climate with a mean annual temperature of 9.3 °C. The mean annual precipitation is 546.8 mm (1954 - 2014), of which precipitation from May to September accounts for 76.9% of the total precipitation. The elevation ranges from 1050 to 1423 m (Xia et al., 2017; Guo et al., 2019). The main landforms include gentle loess-tableland, steep hillslope and gully channel, and their areas account for 57.0%, 15.7% and 27.3%, respectively. The loess-tableland is characterized by low slope (1–5°), gentle and flat terrain and fertile soil. The main soil type is loessial soil with silt loam texture. Most of hillslopes have been constructed as slope-terraces. The main gully channel is usually U-shaped and the branch-gully is more actively developed and easily eroded as a V-shaped by runoff from loess-tableland (Xu et al., 2019). The flat loess-tableland can accumulate the 67.4% of total runoff and cause serious gully erosion that can contribute 86.3% of the total soil erosion (Guo et al., 2019). The original plant species have been seriously destroyed. Since the 1970s, the "Three Protection Belts" system, the "Four Eco-Economical Belts" system and the "Grain for Green" project (Zhao, 1994; Fu et al., 2011) were implemented to control soil erosion. The current mean annual soil erosion rate has been reduced to 4350 t km$^{-2}$ y$^{-1}$ in the study watershed (Guo et al., 2019). The previous vegetation are mainly artificially planted forests and some native secondary herbaceous communities.



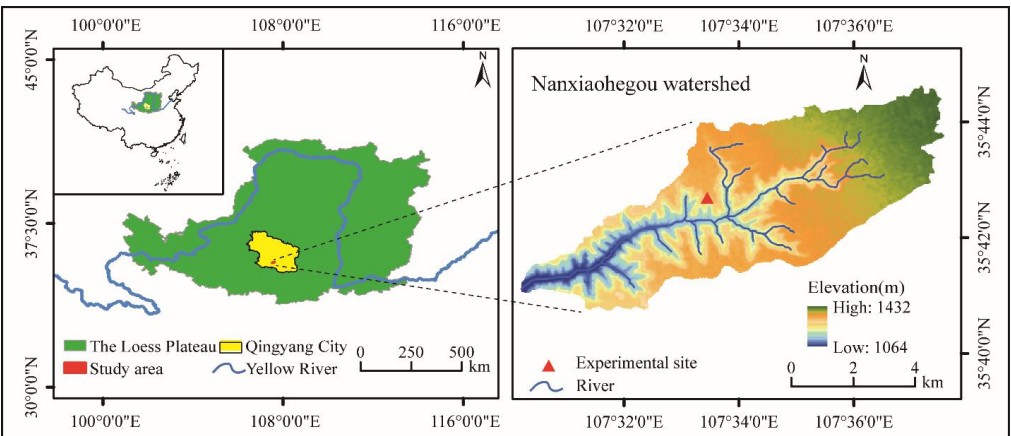


**Figure 1** The location of the experimental site in Nanxiaohegou watershed, Qingyang City, Loess
Plateau, China. Note: The figure production was based on the digital elevation model data (spatial
resolution of 30 m) which is available from http://srtm.csi.cgiar.org (Reuter et al., 2007).

## 2.2 Experimental design
### 2.2.1 Gully head experimental plot construction
Five gully head plots for headcut erosion experiments were constructed at the experimental
station in April 2018. Fig. 2 shows the basic information of the gully head plot consisting of three
landform units (upstream area, headwall and gully bed). The plot width and slope gradient of
upstream area and gully bed are uniformly designed as 1.5 m and 3°, respectively. The upstream area
length, the height of the vertical headwall and the length of the gully bed are 5.0 m long, 0.9 m, and
1.0 m, respectively (Fig. 2a). The plot boundary was constructed in strict accordance with designed
plot dimension using cement and bricks (Fig. 2b). After the construction of plot boundary, the soil
was sieved through a 2 cm sieve with to remove roots and debris and ensure uniform soil underlying
condition. The sieved soil was filled into the plot every 10-cm thick layer according to the
investigated soil bulk density of gully heads. The soil surface of each layer was harrowed to increase
the cohesion between two soil layers (Guo et al., 2019). In general, the filling upstream area length
was 5.5 m that was larger than the precise upstream area length (5.0 m). After establishment of gully
head plots, the five plots were carefully managed about four months (August 2018) to allow the soil
to return to its natural state. During the four-month conservation process, the naturally growing
weeds were weeded out in time. Moreover, a flow-steady tank of 0.6 m, 1.5 m and 0.5 m in length,





width and height was installed at the top of upstream area, and a circular sampling pool of 0.6 m in
diameter was set at the bottom of the gully bed to collect runoff and sediment (Fig. 2a).

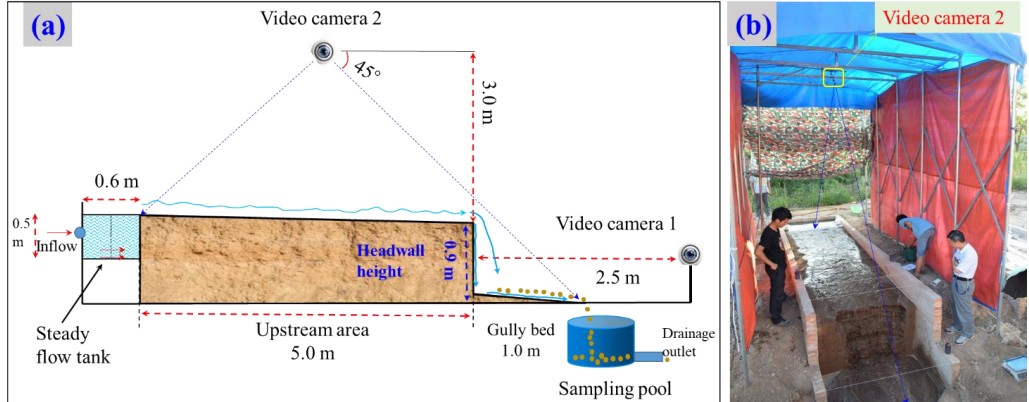


**Figure 2** Sketch (a) and photo (b) of experimental plot.

## 2.2.2 Inflow discharge design

The concentrated runoff generated from upstream area is the main force driving gully headcut
erosion. Jiao et al (1999) concluded that the more serious soil erosion is generally caused by "A"
type rainstorm with the rainfall duration of 25 to 178 mins than other types of rainstorms in the Loess
Plateau. Thus, an extreme case of rainfall duration (180 min) was considered in this study, and the
recurrence period of "A" type rainstorm was designed as 30 years. Previous studies indicated that the
rainstorm distribution on the Loess Plateau showed a non-significant change in past decades (Li et al.,
2010; Sun et al., 2016; Wen et al., 2017). Zhang et al. (1983) proposed a statistical equation (Eq. (1))
for calculating the average rainfall intensity by analyzing 1710 typical rainstorm events in the Loess
Plateau. Then, the inflow discharge was calculated by Eq. (2) and ranged from 3.12 to 9.68 m³ h⁻¹.
Considering the pre-experiment effect, finally, we selected the five inflow discharge levels (3.0, 3.6,
4.8, 6.0, and 7.2 m³ h⁻¹).
$$RI = \frac{5.09N^{0.379}}{(t+1.4)^{0.74}} \quad (1)$$
where $RI$ is the average rainfall intensity during $t$ minutes, mm min⁻¹; $N$ is the recurrence period
of rainstorm, yr; and $t$ is the rainfall duration, min.
$$q = \frac{60\alpha \cdot A \cdot RI \cdot w}{W} \quad (2)$$



where $A$ is the upstream area (km$^2$) and has a wide range of 0.15 - 8.7 km$^2$ according to an early
investigation of research team (Che, 2012); $W$ is the width of the upstream area, km; $w$ is the plot
width, m; and $\alpha$ is the runoff coefficient of bare land and is identified as 0.167 by analyzing the
runoff and rainfall data of standard runoff plots (Li et al., 2006).

## 2.3 Experimental procedure

The scouring experiment was conducted in August 2018. Before formal experiment, firstly, the
upstream area length was adjusted to designed length of 5.0 m (Fig. 3a). Then, a self-made tent
(length × width × height: 6.0 m × 3.0 m × 3.5 m) with waterproof canvas enclosed the plot to resist
the effect of natural rainfall and sunshine on experimental progress and photo shooting for 3D
reconstruction (Fig. 2b). In addition, the experimental process was recorded by two Logitech 930e
video cameras with a resolution of 2.0 megapixels. The camera 1 was installed 2.5 m in front of plot
headwall (Fig. 2a), and the camera 2 was installed 3.0 m above the plot center (Fig. 2a).
Before the experiment, watering can be used to spray each experimental plot until surface runoff
was generated, and then the plot was placed for 24 hours to ensure adequate water infiltration, which
can assure that the soil moisture of the five plots was approximately the same. The inlet pipeline was
placed in steady flow tank when the inflow discharge was adjusted to designed value. A water
thermometer was placed into the steady flow tank to monitor the change in water temperature during
experimental process. The runoff and sediment samples at the plot outlet were collected at 2-min
intervals to represent the temporal change in runoff and sediment of "UA – GH – GB" system, and
the sampling time was also recorded using a stopwatch (Fig. 3b). The runoff and sediment samples
were oven-dried at 105 °C for 24 h and weighed to calculate the soil loss rate of the "UA – GH – GB"
system (g s$^{-1}$). Besides, the timing of the collapse events was also recorded during headcut erosion.
The upstream area was divided into 4 runoff observation sections, and the runoff width ($w$), depth ($d$)
and velocity ($V$) of each section were measured by a calibrated scale of 1 mm accuracy and color
tracer method (Fig. 3b, 3c). The runoff velocity ($V_J$) before runoff arrived at the brink of headcut was
measured 5 – 8 times by the velocity measuring instrument (LS300-A) with the accuracy of 0.01 m
s$^{-1}$, and the runoff width at the headcut brinkpoint was measured (Fig. 3d). The runoff width and
velocity of gully bed were also measured using the same method with upstream area (Fig. 3e). Above





mentioned measurements of runoff characteristics and sediment samples were finished in 2-min
intervals. The whole experimental process was recorded by two video cameras and imported into
computers (Fig. 3f). In addition to above runoff parameters, the runoff depth ($d_b$) at the brink of
headcut, the plunge pool depth ($D_H$) and the vertical distance ($h$) from brink-point of headcut to
water surface of plunge pool were also measured 3 - 5 times by a steel ruler with 1 mm accuracy
within each 2-min intervals (Fig. 4).

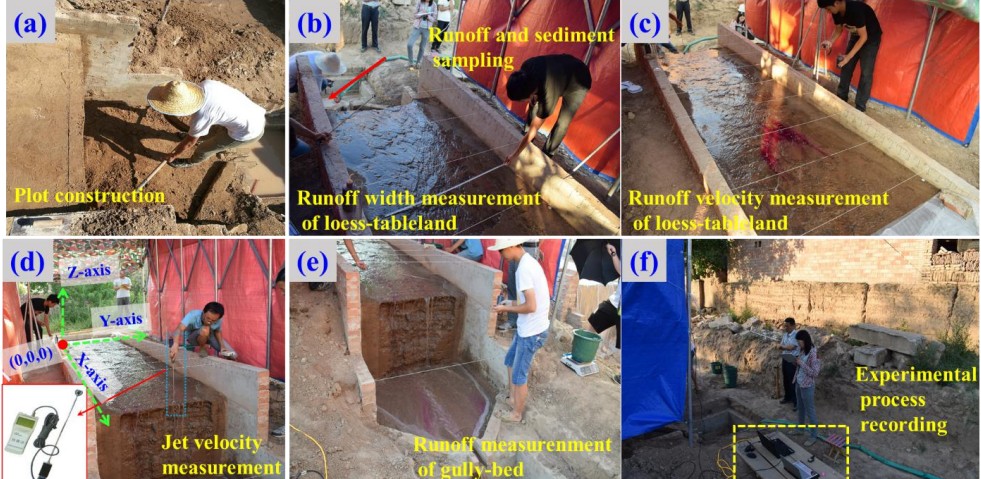


**Figure 3** Runoff and sediment observation and recoding at upstream area, gully head and gully bed.

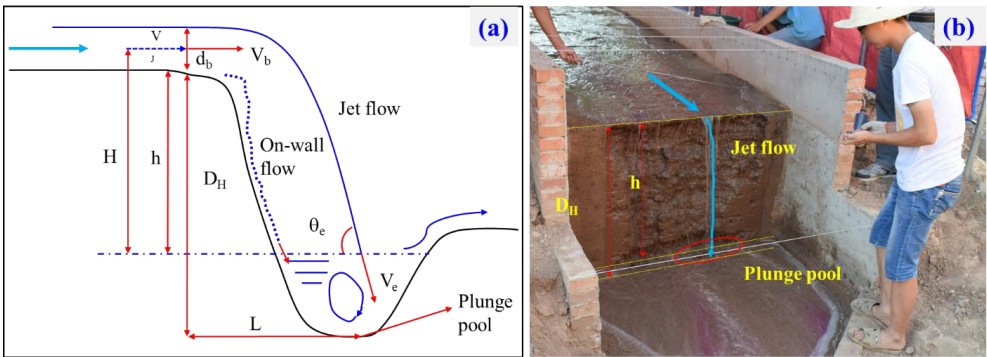


**Figure 4** Sketch of jet flow at gully headcut and plunge pool at gully bed.



To obtain the temporal change in morphological characteristics during gully headcut erosion,
the experimental duration (180 min) was divided into six stage (30 – 60 – 90 – 120 – 150 – 180 min).
Photo-based three-dimensional (3D) reconstruction method was employed to obtain the digital
elevation model data of each plot prior to experiment and after each 30-min test. 14 target points
were placed around the plot for identifying the 3D coordinate before the photos were taken. The
eroded photographic was recorded by a Nikon D5300 camera with the focal length of 50 mm. The
following aspects were required during photos shooting: (1) obvious water on soil surface and direct
sunshine should be avoided, (2) a minimum overlap of 60% between subsequent photographs was
required, and (3) some complex eroded photographic should be taken in detail. In this study, the
upper left corner of the plot was set as the original coordinates (0, 0, 0) , and the direction of
three-dimensional coordinate was determined as shown in Fig. 3d. These collected photos were
imported in Agisoft PhotoScan software (Agisoft LLC, Russia, professional version 1.1.6), and then
these control points and their coordinates would be identified and entered. The root mean square
errors of the target points are 0.0037, 0.0045, 0.0024, 0.0052 and 0.0030 m on average, respectively,
for the experiments of five inflow discharges, which can satisfy the study requirement (millimeter
level). The DEM could be exported and was used to extract the morphological parameters and soil
loss volume of three landform units at six stages (Frankl et al., 2015).
**2.4 Parameter calculation, data analysis and figure plotting**
**2.4.1 Hydraulic parameters of upstream area and gully bed**
Five parameters including runoff velocity ($V$, m s$^{-1}$), Reynold number ($Re$), Froude number ($Fr$),
shear stress ($\tau$, Pa) and stream power ($\omega$, W m$^{-2}$) were used to characterize the changes in hydraulic
properties at upstream area and gully bed positions. The five parameters are calculated as follows.
$$\text{Re} = \frac{V \cdot R}{\upsilon} \quad (1)$$

$$\text{Fr} = \frac{V}{\sqrt{g \cdot R}} \quad (2)$$

$$R = \frac{w \cdot d}{w+2d}, \upsilon = \frac{1.775 \times 10^{-6}}{1+0.0337T+0.000221T^2} \quad (3)$$

$$\tau = \rho_w \cdot g \cdot R \cdot J \quad (4)$$



$$\omega = \tau \cdot V \ (5)$$
where $R$ (m) and $\upsilon$ (m$^2$ s$^{-1}$) are the hydraulic radius and the water kinematic viscosity coefficient,
respectively; $w$ (m), $d$ (m) and $T$ (℃) are the runoff width, depth and water temperature, respectively;
$\rho_w$ (kg m$^{-3}$) is the water density and $J$ (m m$^{-1}$) is the hydraulic gradient.

**2.4.2 Jet properties of gully head**

Based on the measured runoff velocity ($V_J$, m s$^{-1}$) before runoff arrived at the headcut brinkpoint,
the runoff depth ($d_b$, m) at the headcut brinkpoint, the plunge pool depth ($D_H$, m) and the vertical
distance ($h$, m) (Fig. 4a), the three parameters including the runoff velocity at the headcut brinkpoint
($V_b$), jet-flow velocity entry to plunge pool ($V_e$) and jet-flow shear stress ($\tau_j$) were calculated to
clarify the change of jet properties (Rouse, 1950; Hager, 1983; Stein et al., 1993; Flores-Cervantes et
al., 2006; Zhang et al., 2016). The three parameters were calculated as follows.
$$V_b = \begin{cases} \dfrac{\sqrt[3]{q \cdot g}}{0.715}, Fr < 1 \\ V_J \cdot \dfrac{Fr^2 + 0.4}{Fr^2}, Fr > 1 \end{cases} \quad (5)$$
$$Fr = \frac{V_J}{\sqrt{g \cdot d_b}} \quad (6)$$
$$V_e = \frac{V_b}{cos\theta_e}, \theta_e = \arctan\left(\frac{\sqrt{2g \cdot D_H}}{V_b}\right) \quad (7)$$
$$\tau_j = 0.025\left(\upsilon/q\right)^{0.2} \cdot \rho_w \cdot \left[2g \cdot (h + d_b/2) + V_b^2\right] \quad (8)$$

**2.4.3 Energy consumption of upstream area, gully head and gully bed**

In this study, energy consumption of three landform units (upstream area, UA; gully head, GH;
gully bed, GB) were calculated according to the measured runoff characteristic parameters. The
bottom of GB was treated as the zero potential surface to quantify the energy consumption.
Therefore, the total runoff energy ($E_T$, J s$^{-1}$), the runoff energy at the brink of headcut ($E_L$, J s$^{-1}$), the
runoff energy when runoff leaves the plunge pool ($E_H$, J s$^{-1}$), and the runoff energy at the bottom of
gully bed ($E_B$, J s$^{-1}$) were calculated as following.
$$E_T = \rho_w g q\left[\left(L_l + L_g\right)tan\theta + H_h\right] \ (9)$$



$$E_L = \rho_w g q \left[ (L_m + L_g) tan\theta + H_h \right] + \frac{1}{2} \rho_w q V_J^2 \quad (10)$$
$$E_H = \rho_w g q \left( L_m + L_g - V_b \sqrt{\frac{2h}{g}} \right) tan\theta + \frac{1}{2} \rho_w q V_P^2 \quad (11)$$
$$E_B = \frac{1}{2} \rho_w q V_B^2 \quad (12)$$
where the $L_l$ (m) and $L_g$ (m) are the projection length of UA and GB, respectively, during gully
head migration; $L_m$ (m) is the gully head retreat distance; $H_h$ (m) is the initial gully headcut height. $V_P$
(m s⁻¹) and $V_B$ (m s⁻¹) are the runoff velocity runoff leaving the plunge pool and GB, respectively.
Therefore, the total runoff energy consumption ($\Delta E_T$, J s⁻¹), the runoff energy consumption of
UA ($\Delta E_L$, J s⁻¹), the runoff energy consumption of GH ($\Delta E_H$, J s⁻¹) and the runoff energy consumption
of GB ($\Delta E_B$, J s⁻¹) could be calculated as follows.
$$\Delta E_T = E_T - E_B \quad (13)$$
$$\Delta E_L = E_T - E_L \quad (14)$$
$$\Delta E_H = E_L - E_H \quad (15)$$
$$\Delta E_B = E_H - E_B \quad (16)$$

### 291 2.4.4 Statistical analysis

The curve regression analysis method was employed to determine the quantitative relations
between hydraulic characteristics, jet properties, runoff energy consumption and soil erosion rate and
inflow discharge. The fitted equations between soil loss rate of three landform units and hydraulic
characteristics, jet properties, and energy consumption were also quantified by the curve regression.
The soil erosion volume of upstream area, gully head and gully bed were derived from the DEM of
different stages through the ArcGIS 10.0 software. The data analyse was executed by using SPSS
software (version 6.0) and figure plotting was carried out with Origin 8.5 and PowerPoint 2016
software.

### 300 3 Results

### 301 3.1 Spatial-temporal changes in jet properties and runoff hydraulic

### 302 3.1.1 Jet properties of gully head

Fig. 5 shows the temporal variation of three jet property parameters of gully head (GH) under
different inflow discharge conditions. Overall, the flow velocity at the headcut brinkpoint ($V_b$)





increased obviously in the first 30 min and then showed a gradually stable tendency with some
degree of fluctuation (Fig. 5a), and the fluctuation degree was enhanced as the inflow discharge
increased. For example, the $V_b$ increased sharply from 0.66 to 0.88 m s$^{-1}$ during 100 – 124 min under
6.0 m$^3$ h$^{-1}$ inflow discharge due to the headwall failure near headcut enhancing the runoff turbulence.
Regression analysis revealed the significant power relationships ($V_b=a{\cdot}t^b$, $R^2$=0.139-0.704, $P<0.01$)
between $V_b$ and time ($t$) (Table 1). Furthermore, except for 3.6 m$^3$ h$^{-1}$ condition, the $a$-value increased
with the inflow discharge increased, but the $b$-value showed a weak variation (0.08 - 0.10),
indicating that the flow drainage from gully head could improve initial $V_b$ but not change its trend
over time. The mean $V_b$ exhibited a significantly exponential relationship with inflow discharge (Fig.
5b, $P<0.05$). Contrary to the $V_b$, the jet velocity entry to plunge pool ($V_e$) and the jet shear stress ($\tau_j$)
experienced a gradually decreased trend with time (Fig. 5c, 5e). Notably, the $V_e$ and $\tau_j$ suddenly
decreased at 120th min and lasted nearly 40 minutes under 3.0 m$^3$ h$^{-1}$ inflow discharge, which was
mainly attributed to the developed second headcut shortening the jet-flow height. The temporal
change of $V_e$ could be described by logarithmic functions under 3.0 – 4.8 m$^3$ h$^{-1}$ inflow discharge, and
expressed by linear functions under the other inflow discharges, whereas the decrease of the $\tau_j$ with
time could be presented by logarithmic functions under all inflow discharge conditions (Table 1).
Furthermore, both of mean $V_e$ and $\tau_j$ could be expressed by a positive "$S$" function of inflow
discharge (Fig. 5d, 5f).



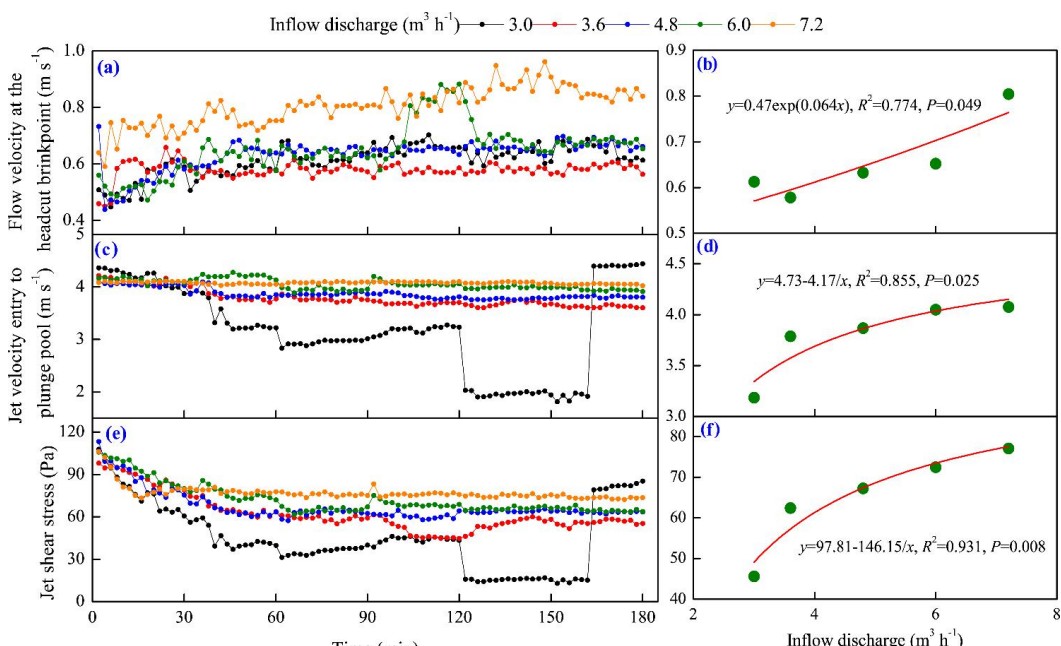

**Figure 5** Temporal changes in jet properties of headcut and their relationships with inflow discharge.

**Table 1** The relationships between jet properties of gully headcut and time.

| Inflow discharge (m³ h⁻¹) | $V_b \sim t$ | $V_e \sim t$ | $\tau_j \sim t$ |
|---|---|---|---|
| 3.0 | $V_b=0.42t^{0.09}$, $R^2=0.691^{**}$ | $V_e=5.28-0.49\lg(t)$, $R^2=0.290^{**}$ | $\tau_j=110.86-15.44\lg(t)$, $R^2=0.344^{**}$ |
| 3.6 | $V_b=0.53t^{0.02}$, $R^2=0.139^{**}$ | $V_e=4.52-0.17\lg(t)$, $R^2=0.859^{**}$ | $\tau_j=117.93-13.14\lg(t)$, $R^2=0.823^{**}$ |
| 4.8 | $V_b=0.46t^{0.08}$, $R^2=0.544^{**}$ | $V_e=4.25-0.09\lg(t)$, $R^2=0.718^{**}$ | $\tau_j=109.22-9.93\lg(t)$, $R^2=0.770^{**}$ |
| 6.0 | $V_b=0.52t^{0.10}$, $R^2=0.509^{**}$ | $V_e=4.17-1.33\times10^{-3}t$, $R^2=0.478^{**}$ | $\tau_j=118.73-10.96\lg(t)$, $R^2=0.876^{**}$ |
| 7.2 | $V_b=0.57t^{0.08}$, $R^2=0.704^{**}$ | $V_e=4.09-1.38\times10^{-4}t$, $R^2=0.111^{**}$ | $\tau_j=95.68-4.42\lg(t)$, $R^2=0.619^{**}$ |

Note: $V_b$, $V_e$ and $\tau_j$ are runoff velocity at the headcut brinkpoint, runoff velocity entry to plunge pool and the jet shear stress, respectively. $^{**}$ refers to the significance of 0.01. The sample number is 90 for the fitted equations.

### 3.1.2 Runoff regime of upstream area and gully bed

The temporal changes in runoff Reynold number ($Re$) and Froude number ($Fr$) of upstream area (UA) and gully bed (GB) and their relationships with inflow discharge are provided in Fig. 6. The $Re$ of UA and GB showed a similar trend over time, that is, the $Re$ firstly increased in the first40 min and then gradually stabilized (Fig. 6a). In addition, the $Re$ of UA was larger than that of GB at any time under same inflow discharge, indicating that the runoff turbulence became weaker after the



runoff of UA passed the gully head. Regression analysis showed the temporal variation in $Re$ of UA
could be described by logarithmic and power functions, but, for the GB, the relationship was mainly
dominated by power function (Table 2). On average, the $Re$ of GB was 50.5% - 65.9% less than that
of UA, and the $Re$ of UA and GB both increased with the increase of inflow discharge as a power
function (Fig. 6b). However, as illustrated in Fig. 6c, the $Fr$ experienced a completely opposite trend
to $Re$. The $Fr$ of UA decreased in the first 60 min and then gradually stabilized, but the $Fr$ of GB
experienced a relatively weak-fluctuating variation over time. For the most of cases, the change in $Fr$
of UA and GB over time could be expressed by logarithmic functions (Table 2). On average, the $Fr$
of UA was 2.39-3.04 times that of GB for same inflow discharge, and the positive power function
could describe the relationship between $Fr$ and inflow discharge (Fig. 6d).

347       Furthermore, the knowledge of open channel hydraulics is adopted to investigate the difference

in runoff regime between UA and GB. The specific definition is: the flow belongs to laminar when
$Re$ is less than 500, the flow is turbulent when $Re$ is larger than 2000, and the flow indicates
transitional when $Re$ ranges from 500 to 2000; and $Fr = 1$ is the critical value for to distinguish the
subcritical and supercritical flow. The six flow regime zones were divided by three boundary lines
($Re = 500$, $Re = 2000$, and $Fr = 1$) according to the logarithmic relationship between the flow
velocity and hydraulic radius (Fig. 7) (Xu et al., 2017b; Guo et al., 2020b). AS shown, the runoff
regimes of UA and GB were located in entirely different zones. The flow of UA was in the
supercritical-transition flow regime in the first 26 min and then gradually transformed to
supercritical-turbulent flow regime under $3.0 - 6.0$ m$^3$ h$^{-1}$ inflow discharge, but the flow at any
moment was in the supercritical-turbulent regime zone under 7.2 m$^3$ h$^{-1}$ inflow discharge. Moreover,
the higher inflow discharge would enhance the flow turbulent degree. The flow of GB belonged to
subcritical-laminar flow category in the initial 6 min, and then transformed to subcritical-transition
and subcritical-turbulent flow regime when inflow discharge was 3.0 and 3.6 m$^3$ h$^{-1}$. The flow was in
the subcritical-turbulent flow regime in most of experimental duration when the inflow discharge is
$4.8 - 7.2$ m$^3$ h$^{-1}$. The difference in flow regime between UA and GB also indicated that the presence
of gully head can greatly reduce flow turbulence.





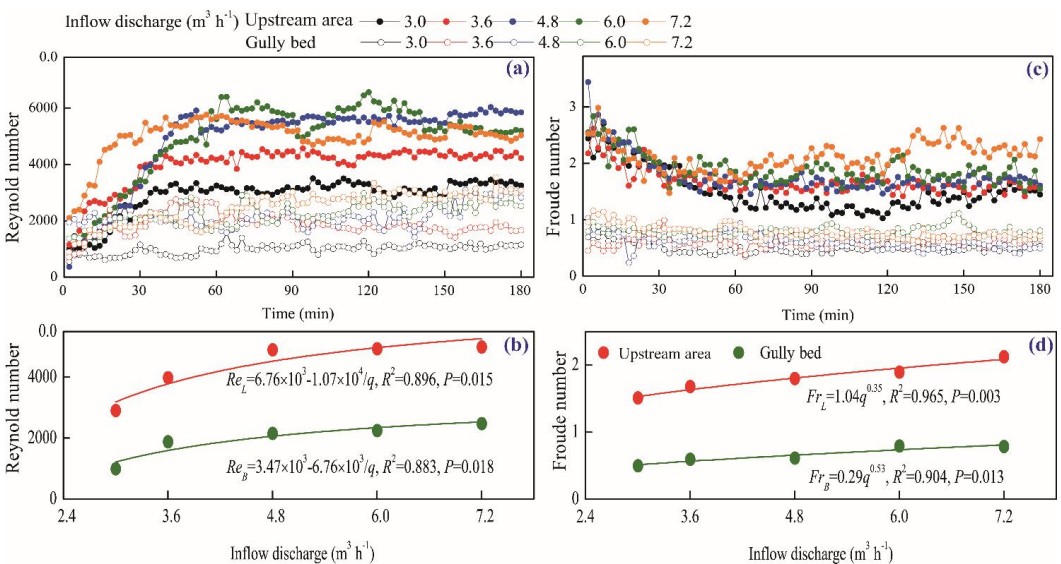

**Figure 6** Temporal changes in runoff regime of upstream area and gully bed and their relationships with inflow discharge.





**Table 2** Relationships between runoff hydraulic parameters and time.

| Variable | Landform unit | Inflow discharge (m³ h⁻¹) | | | | |
|---|---|---|---|---|---|---|
| | | 3.0 | 3.6 | 4.8 | 6.0 | 7.2 |
| Reynold number | UA | $Re=618.69\lg(t)$ $+286.69, R^2=0.761^{**}$ | $Re=705.93\lg(t)$ $+1006, R^2=0.815^{**}$ | $Re=1433\lg(t)$ $-1159, R^2=0.849^{**}$ | $Re=946.64t^{0.38}$, $R^2=0.794^{**}$ | $Re=2760t^{0.14}$, $R^2=0.486^{**}$ |
| | GB | $Re=514.36t^{0.15}$, $R^2=0.504^{**}$ | — | $Re=4.31t+1760$, $R^2=0.334^{**}$ | $Re=1.12\times10^3t^{0.16}$, $R^2=0.566^{**}$ | $Re=744.99t^{0.28}$, $R^2=0.872^{**}$ |
| Froude number | UA | $Fr=2.89-0.33\lg(t)$, $R^2=0.651^{**}$ | $Fr=2.46-0.19\lg(t)$, $R^2=0.651^{**}$ | $Fr=3.27-0.35\lg(t)$, $R^2=0.656^{**}$ | $Fr=2.76-0.20\lg(t)$, $R^2=0.515^{**}$ | — |
| | GB | $Fr=0.72-0.05\lg(t)$, $R^2=0.326^{**}$ | — | $Fr=1.0-0.09\lg(t)$, $R^2=0.359^{**}$ | — | $Fr=1.21-0.10\lg(t)$, $R^2=0.634^{**}$ |
| Shear stress | UA | $\tau=0.66\lg(t)+0.55$, $R^2=0.737^{**}$ | $\tau=1.18\lg(t)+0.78$, $R^2=0.813^{**}$ | $\tau=1.32\lg(t)-0.62$, $R^2=0.817^{**}$ | $\tau=1.50\lg(t)-0.63$, $R^2=0.663^{**}$ | $\tau=1.11\lg(t)+0.99$, $R^2=0.819^{**}$ |
| | GB | $\tau=2.44t^{0.08}$, $R^2=0.205^{**}$ | $\tau=3.88t^{0.05}$, $R^2=0.106^{**}$ | $\tau=2.27t^{0.19}$, $R^2=0.664^{**}$ | $\tau=3.64t^{0.12}$, $R^2=0.212^{**}$ | $\tau=1.99t^{0.27}$, $R^2=0.686^{**}$ |
| Stream power | UA | $\omega=0.34\lg(t)+0.16$, $R^2=0.761^{**}$ | $\omega=0.38\lg(t)+0.55$, $R^2=0.815^{**}$ | $\omega=0.78\lg(t)-0.63$, $R^2=0.849^{**}$ | $\omega=0.69\lg(t)-0.23$, $R^2=0.737^{**}$ | $\omega=0.27\lg(t)+1.56$, $R^2=0.436^{**}$ |
| | GB | $\omega=0.28t^{0.15}$, $R^2=0.504^{**}$ | $\omega=0.69t^{0.09}$, $R^2=0.123^{**}$ | $\omega=0.50t^{0.19}$, $R^2=0.540^{**}$ | $\omega=0.83t^{0.09}$, $R^2=0.338^{**}$ | $\omega=0.51t^{0.23}$, $R^2=0.806^{**}$ |

Note: UA and GB refer to upstream area and gully bed. $Re$, $Fr$, $\tau$ and $\omega$ are Reynold number, Froude number, shear
stress, stream power, respectively. $^{**}$ refers to the significance of 0.01. The sample number is 90 for the fitted
equations.

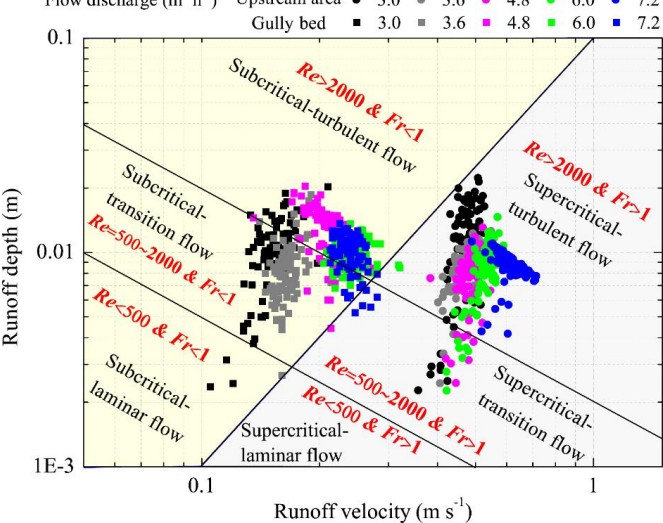

**Figure 7** Runoff regime zones of upstream area and gully bed under different inflow discharge conditions.
**3.1.3 Runoff shear stress and stream power of upstream area and gully bed**
Fig.8 shows the temporal changes in runoff shear stress ($\tau$) and stream power ($\omega$) of upstream



area (UA) and gully bed (GB) and their relationships with inflow discharge. Overall, the τ of UA and
GB exhibited a gradually increased trend in the first 60 min, and whereafter, a relative steady state
was obtained, but the larger inflow discharge perturbed the steady situation (Fig. 8a). Furthermore,
the temporal change in τ of UA could be expressed by logarithmic functions, and the τ of GB
showed a significant power function with experimental time (Table 2). On average, the τ of GB was
2.8% - 15.7% larger than the UA. The averaged τ of UA and GB increased with inflow discharge as a
power function ($\tau = a - b/q$), and the GB had a faster increased-speed ($b$-value) than UA (Fig. 8b),
signifying that the difference in τ between UA and GB would be widened with the inflow discharge
increased. Similarly, the ω of UA and GB also exhibited a trend of gradual increase and stabilization
over time (Fig. 8c). Different from the temporal change in τ, the ω of GB was always less than that
of UA at any time for all inflow discharge conditions. Likewise, the variation in ω of UA and GB
over time exhibited a significant logarithmic and power function, respectively. On average, the ω of
GB was 49.2% - 65.9% less than UA, and the positive increase in ω of UA and GB with the inflow
discharge could be expressed by a power function (Fig. 8d).

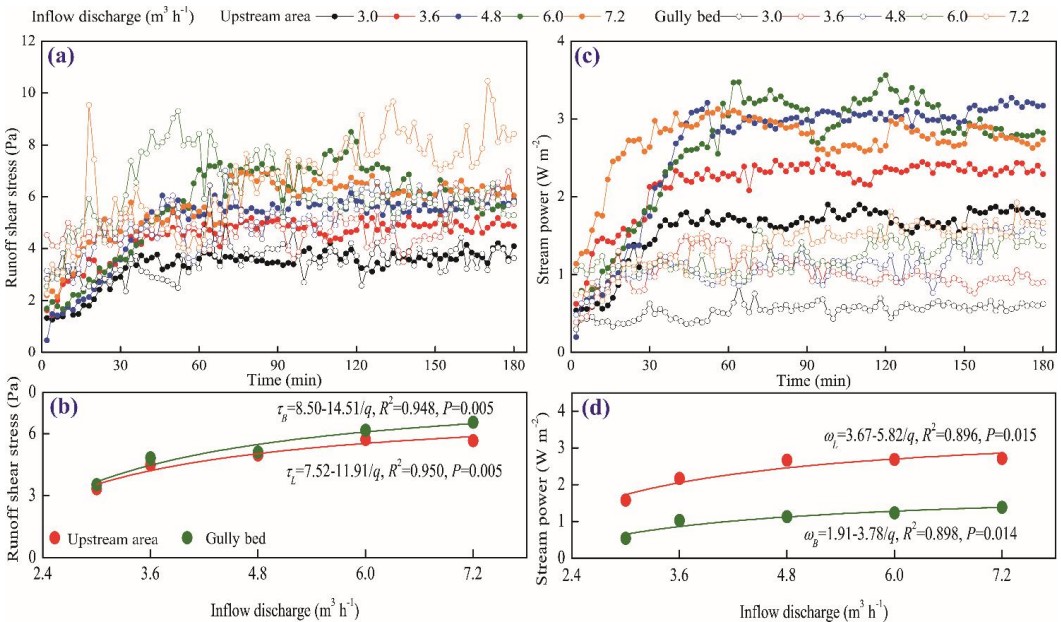


**Figure 8** Temporal changes in runoff shear stress and stream power of upstream area and gully bed and their
relationships with inflow discharge


## 3.2 Spatial-temporal change of energy consumption


Fig. 9 illustrates the temporal change in accumulated energy consumption of upstream area
(UA), gully head (GH) and gully bed (GB). The accumulated energy consumption of the three
landform units continued to linearly increase with time ($R^2$=0.990-0.999, $P<0.01$), of which the
accumulated energy consumption in GH was always the highest at any time, followed by UA and GB
for the experiments of five inflow discharges. Moreover, the energy consumption rate (the
slope-value of fitted equation) in the three landform units is basically constant, indicating the
spatial-temporal change in energy consumption maintained a relatively steady state during gully
headcut erosion. Moreover, the energy consumption rate of GH was the highest, followed by UA and
GB, and the energy consumption rate in the three landform units also increased with the increase of
inflow discharge.
The variations of total energy consumption of UA, GH and GB and their proportions with
inflow discharge are shown in Fig. 10. As illustrated in Fig. 10a, both of the total energy
consumption of the "UA-GH-GB" system and the three landform units increased with the increase of
inflow discharge. When inflow discharge increased from 3.0 to 7.2 m$^3$ h$^{-1}$, the total energy
consumption of the system, UA, GH and GB increased by 3.6% - 105.3%, 3.4% - 62.0%, 3.5% -
108.2% and 9.0% - 327.5%, respectively. Regression analysis revealed that the energy consumption
of system and the three landform units increased with inflow discharge as an exponential function
($y=a\cdot\exp(b\cdot x)$, $a$=1.14 - 55.41, $b$=0.13 - 0.36, $R^2$=0.954 - 0.992, $P<0.05$). Furthermore, in view of the
proportion of energy consumption, the energy consumption of UA accounted for 15.6% - 19.8% of
total energy consumption, and linearly decreased with inflow discharge increased ($R^2$=0.933,
$P<0.05$), whereas the proportion in GB (2.8% - 5.8%) linearly increased with inflow discharge
increased ($R^2$=0.983, $P<0.05$). However, the proportion of energy consumption (77.3% - 78.6%) in
GH showed a weak change with inflow discharge (Fig. 10b), signifying that the most of runoff
energy (77.5% on average) was consumed in the gully head position during headcut migration.





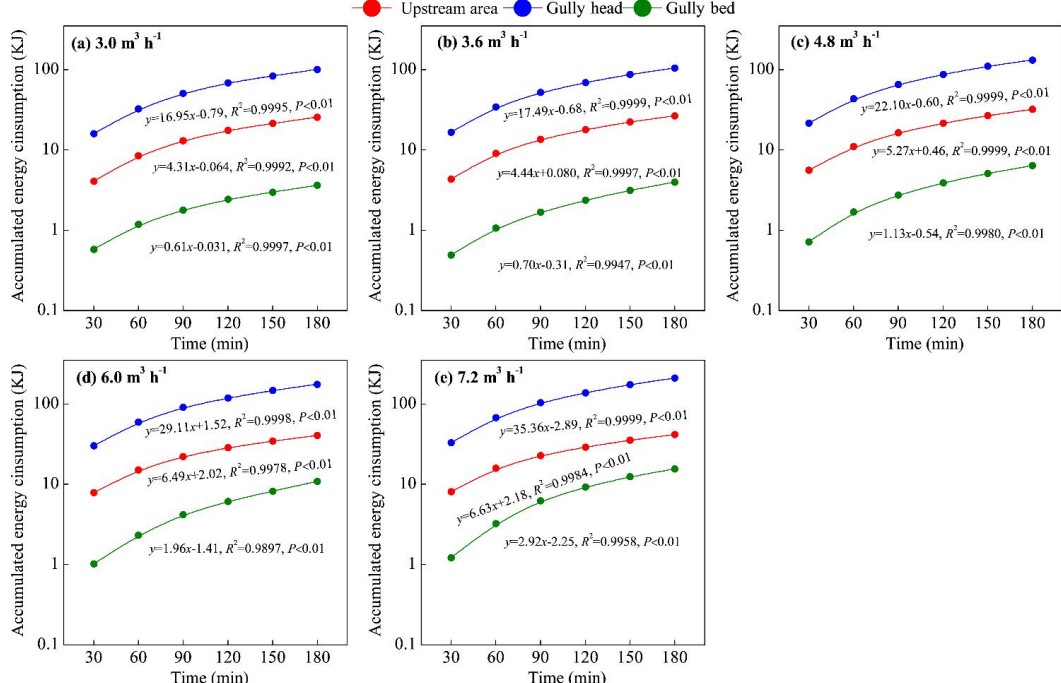

**Figure 9** Temporal changes in runoff energy consumption of upstream area, gully head and gully bed under different inflow discharge conditions

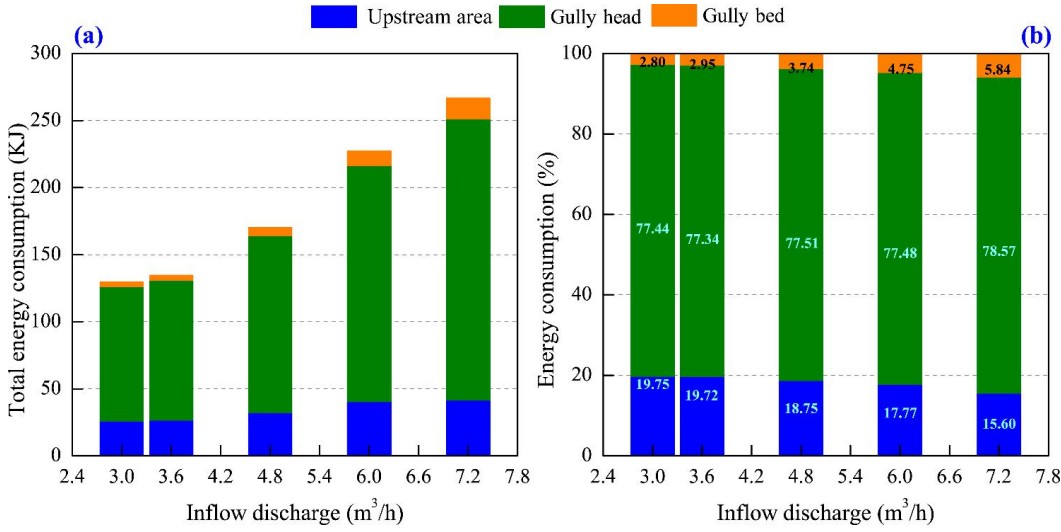

**Figure10** The variation in energy consumption of upstream area, gully head and gully bed and their proportions with inflow discharge





### 3.3 Spatial-temporal change of soil loss

### 3.3.1 Soil loss process

Fig. 11a shows that the soil loss rate of the "upstream area (UA)—gully head (GH)—gully bed (GB)" system rose to a peak in first 20 min, then gradually descend and levelled off. Especially for the 6.0 and 7.2 $m^3$ $h^{-1}$, the soil loss rate showed a severe fluctuation trend in the first 30 min. The peak soil loss rate increased from 75.4 to 306.9 g $s^{-1}$ with increasing inflow discharge. The soil loss of UA and GH experienced a similar change process. The soil loss rate was the highest in the early stage of the experiment, and gradually decreased with time, and became stable after 120 min (Fig. 11b, 11c). Furthermore, the temporal variation in soil loss of UA and GH could be well expressed by logarithmic function ($S_L=a-b\cdot\ln(t)$, $P<0.05$, Table 3), and the $a$-value (representing initial soil loss rate) and $b$-value (reflecting the reduction rate of soil loss rate with time) increased with increasing inflow discharge, indicating that larger inflow discharge can improve initial soil loss of UA and GH and also expedite the decrease of soil loss rate.

However, the GB presented a completely different soil loss process from UA and GH (Fig. 11d). The GB was always characterized by sediment deposition during the whole experiment for the 3.0 – 4.8 $m^3$ $h^{-1}$ inflow discharges. The sediment deposition rate gradually decreased with time and presented a significant "S" function over time ($S_B=a/t-b$, $R^2=0.918-0.982$, $P<0.01$, Table 3). When the inflow discharge was larger than 4.8 $m^3$ $h^{-1}$, the sediment generated from UA and GH was deposited firstly in the GB and then gradually transported, and the temporal change of deposited sediment on GB accorded with logarithmic functions ($R^2=0.936$ and $0.906$, $P<0.01$, Table 3). Furthermore, two critical time points (135 min and 111 min) can be derived from the two fitted logarithmic equations, which distinguished sediment deposition from sediment transport, signifying that the runoff began to transport the sediment deposited on GB after 135 min and 111 min for 6.0 and 7.2 $m^3$ $h^{-1}$ inflow discharge.



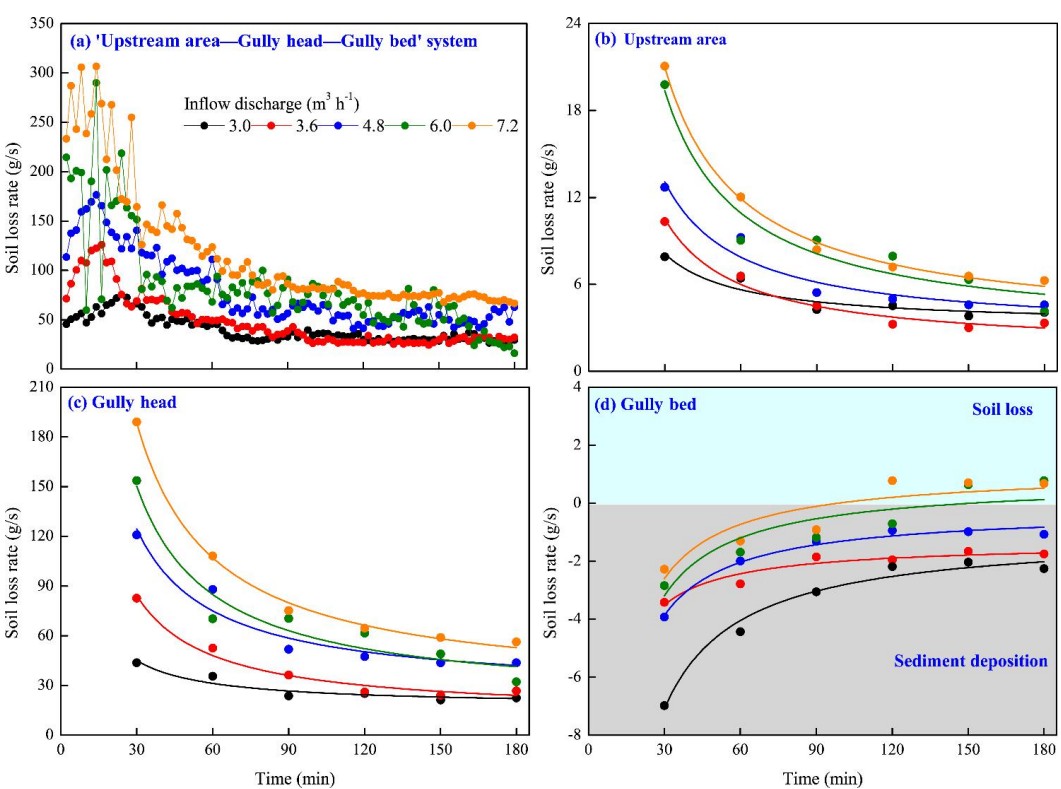

**Figure 11** Temporal variation in soil loss rate of the "upstream area—gully head—gully bed" system and each landform unit

**Table 3** Relationships between soil loss rate of three landform units and time

| Inflow discharge ($m^3 h^{-1}$) | Fitted equations | | |
|---|---|---|---|
| | Upstream area | Gully head | Gully bed |
| 3.0 | $S_L=15.71-2.34\ln(t)$, $R^2=0.909^{**}$ | $S_H=87.12-12.99\ln(t)$, $R^2=0.908^{**}$ | $S_B=-182.62/t-1.01$, $R^2=0.980^{**}$ |
| 3.6 | $S_L=23.97-4.18\ln(t)$, $R^2=0.938^{**}$ | $S_H=191.82-33.44\ln(t)$, $R^2=0.939^{**}$ | $S_B=-64.46/t-1.36$, $R^2=0.918^{**}$ |
| 4.8 | $S_L=28.76-4.85\ln(t)$, $R^2=0.930^{**}$ | $S_H=273.64-46.17\ln(t)$, $R^2=0.929^{**}$ | $S_B=-109.36/t-0.22$, $R^2=0.982^{**}$ |
| 6.0 | $S_L=44.0-7.69\ln(t)$, $R^2=0.884^{*}$ | $S_H=341.59-59.74\ln(t)$, $R^2=0.885^{*}$ | $S_B=2.03\ln(t)-9.96$, $R^2=0.936^{**}$ |
| 7.2 | $S_L=47.34-8.25\ln(t)$, $R^2=0.922^{**}$ | $S_H=425.24-74.07\ln(t)$, $R^2=0.924^{**}$ | $S_B=1.86\ln(t)-8.76$, $R^2=0.906^{**}$ |

Note: $S_L$, $S_H$ and $S_B$ are the soil loss rate of upstream area, gully head and gully bed, respectively. The sample No. is 6 for fitting equation. $^{*}$ and $^{**}$ indicate the significant level of 0.05 and 0.01.

## 3.3.2 Spatial distribution of soil loss

The variation in soil loss amount and proportion of the three landform units (UA, GH, GB) with inflow discharge is shown in Fig. 12. As illustrated in Fig. 12a, for the experiments of five inflow discharges, the soil loss was dominant in the UA and GH, but the GB was dominated by sediment





deposition due to the weaker sediment transport capacity of runoff on GB than sediment
deliverability of UA and GH. Furthermore, the soil loss amount of UA and GH ranged from 55.9 to
110.7 kg and from 310.0 to 994.8 kg, respectively, and increased linearly with increasing inflow
discharge ($R^2$=0.966 and 0.969, $P$<0.05). The sediment deposition amount of GB ranged from 4.2 to
37.7 kg, and decreased with inflow discharge as a logarithmic function ($R^2$=0.961, $P$<0.05). In terms
of proportion of soil loss (Fig. 12b), the proportion of UA and GH reached the maximum (15.3%)
and minimum (84.7%), respectively under 3.0 m$^3$ h$^{-1}$ inflow discharge, whereas, the proportion
exhibited a little change (UA: 9.5% - 11.4%; GH: 88.6% - 90.5%) when the inflow discharge is 7.2
m$^3$ h$^{-1}$. Remarkably, the proportion of deposited sediment amount on GB to total soil loss amount
ranged from 0.4% to 10.3%, and decreased exponentially with inflow discharge ($R^2$=0.992,
$P$<0.001).

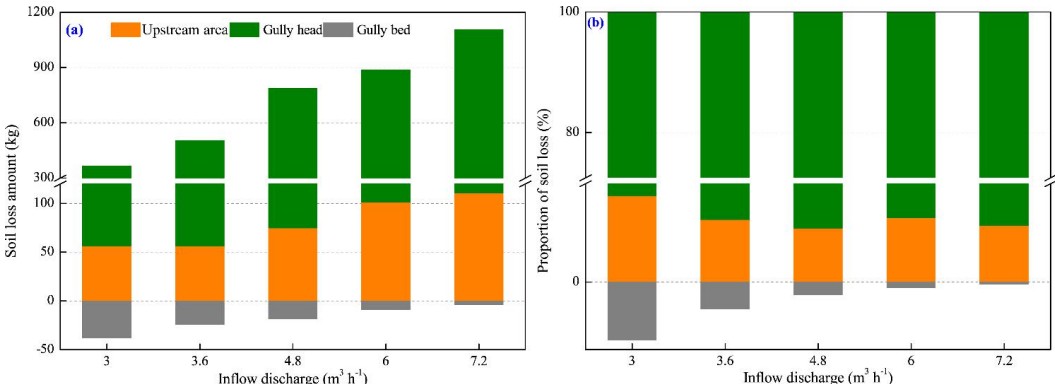

**Figure 12** Variation in soil loss amount and proportion of upstream area, gully head and gully bed with inflow
discharge
**3.4 Spatial change in hydrodynamic mechanism of soil loss**
**3.4.1 Relationships between soil loss and hydraulic parameters**
Fig. 13 indicates the significant difference in the relationships between soil loss rate and
hydraulic parameters among the three landform units (Fig. 13). For the upstream area (UA), the soil
loss rate could be described as a series of exponential functions of runoff velocity, Reynold number,
Froude number, runoff shear stress and stream power, of which the runoff shear stress and stream
power showed a closer correlation with soil loss (Fig. 13a - 13e, $R^2$=0.830 – 0.945). Furthermore, the





increased speed of soil loss rate obviously increased with the increasing hydraulic parameters (except
for runoff velocity), indicating that soil loss of UA showed a stronger sensitive response to increasing
hydraulic properties. However, the soil loss rate of gully bed (GB) linearly increased with the
above-mentioned five parameters (Fig. 13f – 13j, $R^2$=0.918 – 0.994), which suggested that the
decreased rate of sediment deposition of GB is basically constant with the increasing hydraulic
properties. Further analysis showed that there are critical runoff velocity, Reynold number, Froude
number, runoff shear stress and stream power for triggering the transformation of sediment
deposition to soil erosion on GB, and the critical values are 0.26 m s$^{-1}$, 2845, 0.85, 6.94 Pa and 0.40
W m$^{-2}$, respectively. For the gully head (GH) position, the soil loss was significantly affected by jet
velocity entry to plunge pool and jet shear stress (Fig. 13l and 13m, $R^2$=0.862 and 0.939), while the
relationship between soil loss and flow velocity at the headcut brink-point was not significant (Fig.
13k, $P$=0.065).



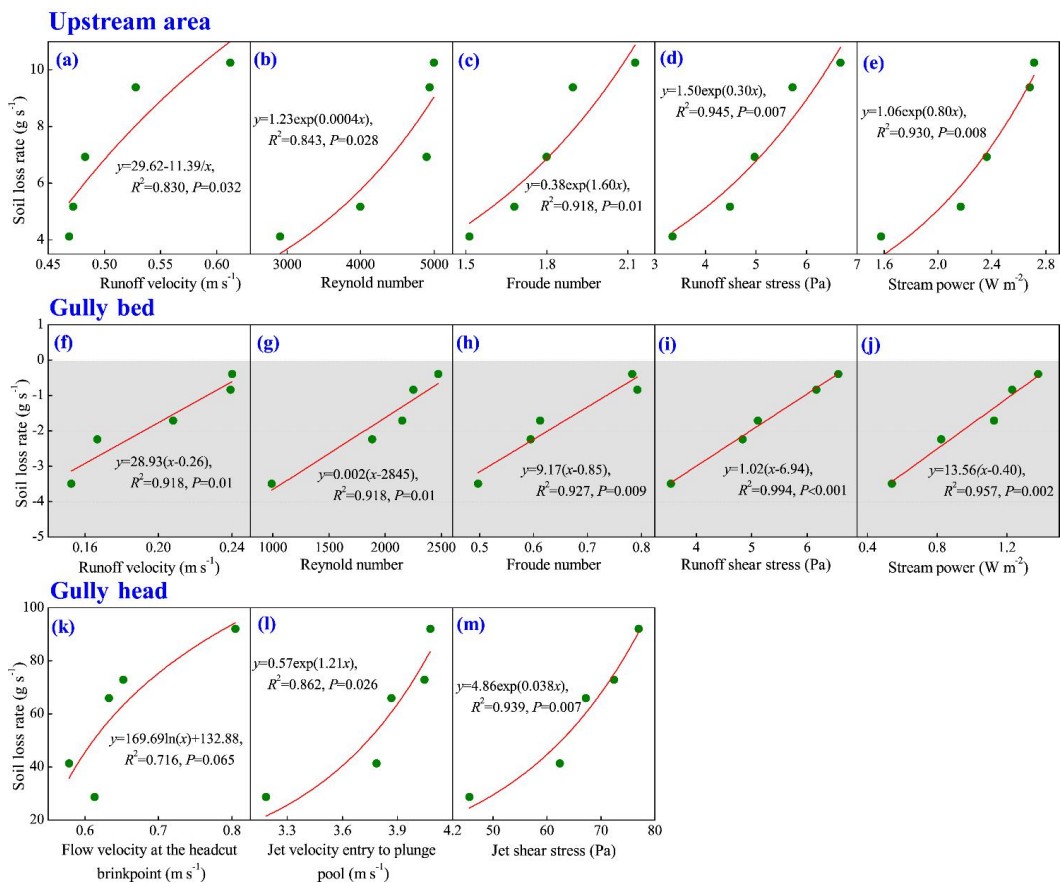

**Figure 13** Relationships between soil loss rate of three landform units and hydraulic and jet properties

## 3.4.2 Response of soil loss to energy consumption

As illustrated in Fig. 14, the soil loss rate of three landform units was positively and significantly related to the energy consumption ($P<0.05$), and a logarithmic function was found to fit the relationship between soil loss rate and energy consumption best ($R^2=0.889 – 0.987$). Furthermore, there is critical energy consumption to initiate soil erosion of the upstream area (UA) and gully head (GH) based on the fitted logarithmic functions (Fig. 14a, b). The critical energy consumption for GH (5.79 J s$^{-1}$) is 2.57 times greater than that (1.62 J s$^{-1}$) of the UA. Similarly, for the gully bed (Fig. 14c), the minimum energy consumption (1.64 J s$^{-1}$) is needed to trigger the transformation of sediment deposition to soil loss.



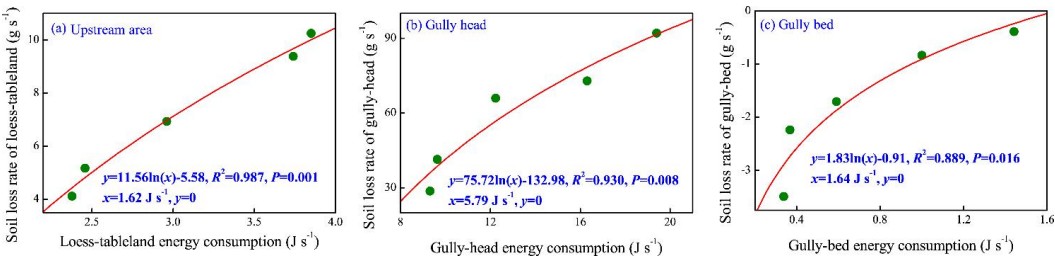

**Figure 14** Relationships between soil loss rate of three landform units and runoff energy consumption

## 4 Discussion

### 4.1 Spatial-temporal changes in hydraulic properties

This study revealed that the runoff velocity at the headcut brink-point ($V_b$) firstly raised and then gradually stabilized with time (Fig. 5a), which was closely corresponded to the gradually decreased runoff width on the upstream area with time (Shi et al., 2020a). However, this result was inconsistent with Zhang et al (2016, 2018) and Shi et al (2020b) who reported that the $V_b$ decreased over time, which was mainly due to the gradually increased roughness and resistance of underlying surface over time reducing the runoff velocity (Su et al., 2015). The further analysis of power function between $V_b$ and time ($V_b=a·t^b$, Table 1) showed that the $a$-value increased but the $b$-value showed a weak variation with the inflow discharge increased, indicating that upstream flow can improve initial $V_b$ but not affect its change trend over time. By contrast, the jet velocity entry to plunge pool ($V_e$) and jet shear stress ($\tau_j$) experienced a gradually decreased process (Fig. 5c, 5e), which was mainly attributed to the shortening of jet-flow height caused by the development of second headcut and upstream flow undercutting headcut brink-point (Guo et al., 2019). This result, however, differed from the finding of Zhang et al. (2016) who stated the $V_e$ and $\tau_j$ remained stable as the experiments progressed, which was mainly attributed to the weak change of jet-flow height induced by slow headcut retreat.

For the runoff hydraulic of upstream area (UA) and gully bed (GB), the Reynold number $Re$ of UA and GB initially increased and gradually stabilized, but the Froude number $Fr$ showed an opposite trend. This phenomenon was agreed with previous studies (e.g. Su et al., 2015; Zhang et al., 2016; Shi et al., 2020a). Besides, for the same upstream inflow discharge, the $Re$ and $Fr$ of UA were larger than that of GB by 50.5%-65.9% and 1.39-2.04 times, respectively, indicating that the runoff



turbulence became weaker after the runoff of UA passed the gully head and plunge pool. More

evidently, the runoff on UA was in the supercritical-transition and supercritical-turbulent flow regime

(Re > 500, Fr > 1), whereas the runoff on GB belonged to subcritical-transition and

subcritical-turbulent flow regime (Re > 500, Fr < 1). The above result was supported by Shi et al.

(2020a) who stated that the $Re$ of gully bed decreased by 1.5%-30% as the flow fell from the

upstream area, but Su et al. (2015) suggested that the steady state $Re$ of gully bed was higher than

that of upstream area. In the study of Su et al. (2015), the larger gully bed slope gradient than

upstream area would accelerate the runoff velocity and thus enhance flow turbulence (Bennett, 1999;

Pan et al., 2016). Our study found that temporal variation in the shear stress (τ) and stream power (ω)

of UA was similar with GB, and, compared to UA, the τ and ω of GB increased and decreased by

2.8% - 15.7% and 49.2% - 65.9%, respectively. This was different from some previous experimental

studies on gully and bank gully. For example, the result from the study of Shi et al. (2020a) indicated

that the τ of gully bed decreased by 65.9% - 67.1%, compared to catchment area, and a similar result

was also found during bank gully headcut erosion (Su et al., 2015). Previous studies also have

proven that the change in hydraulic properties from upstream area to gully bed is affected by various

factors including plunge pool size, slope gradient, initial step height, and soil texture (Bennett and

Casalí, 2001; Wells et al., 2009a, 2009b).

## 4.2 Spatial-temporal change in runoff energy consumption and soil erosion

Our study revealed that the accumulated runoff energy consumption of the upstream area (UA),

gully headcut (GH) and gully bed (GB) linearly increased over time (Fig. 9), indicating the

spatial-temporal change in energy consumption maintained a relatively steady state during gully

headcut erosion. However, the flow energy consumption of bank gully in three landform units

logarithmically increased over time (Su et al., 2015). This difference further manifested that the

runoff energy consumption of different landform units depends on gully type to some extent as well

as soil texture, slope and headwall height (Wells et al., 2009a). Besides, under this flow discharge

conditions, the proportion of energy consumption in UA, GH and GB was 15.6%-19.8%,

77.3%-78.6% and 2.8%-5.8%, respectively (Fig. 10), which was also indirectly supported by the

study of Su et al. (2015) who suggested that the runoff energy consumption per unit soil loss from





upstream area, headcut and gully bed is 17.4%, 70.5% and 12.0%, respectively. This further signified
that the gully head consumed the most of runoff energy (77.5% on average) during headcut
migration. The flow energy must be consumed to surmount the soil resistance as headcut migrates,
and the consumed energy was mainly focused on headwall and plunge pool development (Alonso et
al., 2002).

564        In terms of soil loss, our study indicated that the soil loss rate of the "UA-GH-GB" system

initially increased to the peak value and then gradually declined and stabilized (Fig. 11), which was
consistent with the results of many studies on rill and gully headcut erosion under different
conditions (slope, initial step height, flow discharge, soil type, soil stratification) (Bennett, 1999;
Bennett and Casalí, 2001; Gordon et al., 2007; Wells et al., 2009a; Shi et al., 2020a). Both the scour
depth and sediment production increased in the initial period of underlying surface adjustment, while
once the plunge pool development was maintained, and sediment yield decreased and gradually
stabilized (Bennett et al., 2000). In addition, the significant difference in soil loss process was found
among the three landform units. The soil loss of UA and GH decreased logarithmically over time,
which was similar with several studies (e.g. Su et al., 2015; Shi et al., 2020b). Nevertheless, the GB
was always characterized by sediment deposition for the inflow discharge of < 4.8 m$^3$ h$^{-1}$, whereas
the sediment was deposited firstly and then gradually transported as the inflow discharge increased to
6.0 and 7.2 m$^3$ h$^{-1}$. Similar results were also found in some previous studies on rill heacut erosion
(Bennett, 1999; Bennett and Casalí, 2001; Gordon et al., 2007; Wells et al., 2009a). However, Su et
al. (2014, 2015) revealed a larger soil loss volume or soil loss rate in gully bed than upstream area
and headwall during bank gully headcut erosion. This difference between our study and Su et al.
(2014, 2015) is primarily caused by the difference in slope gradient. The gully bed slope (20°) of
bank gully was larger than that (3 °) of our study, indicating the runoff on gully bed of bank gully
had stronger sediment transport capacity (Zhang et al., 2009; Ali et a., 2013; Wu et al., 2016, 2018).
In view of the proportion of soil loss, the proportion of UA and GH was 9.5% - 11.4% and 88.6% -
90.5%, respectively, of which the proportion of deposited sediment on GB to the sediment yield from
UA and GH can reach up to 0.4% - 10.3%. This result fully demonstrated that the gully head is the
main source of sediment production during gully headcut erosion (Oostwoud-Wijdenes & Bryan,





1994; Zhao, 1994; Su et al., 2014), and also manifested the necessary and importance of gully
headcut erosion controlling in gully-dominated region.

## 4.3 Hydrodynamic characteristics of headcut erosion

The significant different relationships between soil loss and jet or hydraulic characteristics was
found among UA, GH, and GB. The soil loss rate of UA exponentially increased with five hydraulic
parameters (runoff velocity, Reynold number, Froude number, runoff shear stress and stream power),
indicating that soil loss of UA showed a stronger sensitive response to increasing hydraulic
properties. This could attribute to the frequent bank collapse on UA accelerating soil loss (Wells et
al., 2013; Qin et al., 2018). However, the sediment deposition rate of GB linearly decreased with the
five hydraulic parameters, signifying that sediment deposition on GB decreased at a stable state with
the increase of hydraulic parameters. Therefore, the sediment deposition rate would reach zero when
the five hydraulic parameters increased to the critical values, implying that the transformation of
sediment deposition to sediment transport on GB would be triggered. Furthermore, the shear stress is
the optimal parameter describing soil loss process of UA and GB, which differed from some studies
on hillslope erosion hydrodynamic characteristics (Zhang et al., 2009; Shen et al., 2019; Ma et al.,
2020). Most of studies have verified that stream power is the superior hydrodynamic parameter
describing soil detachment process. This comparison also fully illustrated the great difference in
hydrodynamic characteristic between hillslope erosion and headcut erosion. In this study, the soil
loss of gully head (including plunge pool erosion) was significantly affected by jet properties. It's
confirmed that the plunge pool erosion by jet flow becomes a crucial process controlling gully head
migration and sediment production (Oostwoud-Wijdenes et al., 2000). Consequently, the plunge pool
erosion theory is usually employed to build several headcut retreat models (Alonso et al., 2002;
Campo-Bescós et al., 2013). Although the weak correlation between soil loss of gully head and flow
velocity at headcut breakpoint, the larger flow velocity resulted from increasing inflow discharge
would improve the shear stress of jet flow impinging gully bed, and thus the gully headcut suffered
stronger incisional erosion of the plunge pool. However, in fact, the soil loss of gully head was also
affected by on-wall flow erosion (Chen et al., 2013), and thus more studies should be conducted to
clear the effect of on-wall flow properties on headcut erosion.




From the energy consumption perspective, the soil loss rate of the three landform units
significantly and logarithmically increased with the energy consumption, and the similar change
trend was also found in the study of Su et al. (2015). This finding suggests that energy consumption
could be considered as the available parameter to estimate the soil loss of gully headcut erosion (Shi
et al., 2020b). Furthermore, we found the critical energy consumption initiating soil erosion of UA,
GH, and GB are 1.62 J s$^{-1}$, 5.79 J s$^{-1}$ and 1.64 J s$^{-1}$, respectively, indicating the soil loss of gully head
(including plunge pool) needs more flow energy consumption (Zhang et al., 2018; Shi et al., 2020a,
2020b). This phenomenon can be attributed to the fact that the more runoff energy was consumed at
the gully headwall and plunge pool erosion than UA and GB and thus resulted in more severe soil
loss during headcut erosion.
## Summary
This study investigated the temporal-spatial changes in flow hydraulic, energy consumption and
soil loss during headcut erosion based on a series of scouring experiments of gully headcut erosion.
The jet properties of gully head (GH) were significantly affected by upstream flow discharge. The
upstream area (UA) and gully bed (GB) had similar temporal changes in Reynold number, Froude
number, shear stress and stream power. The flow was supercritical on UA, but subcritical on GB, and
the turbulent degree was enhanced by the increasing inflow discharge. The flow Reynold number,
shear stress and stream power decreased by 56.0%, 63.8% and 55.9%, respectively, but Froude
number increased by 7.9% when flow passed the gully headcut and plunge pool. The accumulated
energy consumption at UA, GH and GB linearly increased with time, of which the GH consumed
77.5% of the total runoff energy. The soil loss of UA and GH decreased logarithmically over time,
whereas the GB was mainly characterized by sediment deposition. The GH can contribute 88.5% of
total soil loss, of which 3.8% sediment production was deposited on GB. The soil loss of UA and GH
and the sediment deposition of GB were significantly affected by hydraulic and jet properties. Our
study revealed that the critical energy consumption to initiate soil erosion of UA, GH and GB are
1.62 J s$^{-1}$, 5.79 J s$^{-1}$ and 1.64 J s$^{-1}$, respectively. The runoff energy consumption could be considered
as a non-negligible parameter to predict soil loss of gully headcut erosion.



## Data availability

At present, the data are not publicly accessible because of a situation that we don't have permission to share data according to the requirement of the funded program and our institute.

## Author contribution

Mingming Guo and Wenlong Wang designed the experiments. Mingming Guo, Zhuoxin Chen, Tianchao Wang, Qianhua Shi, Man Zhao and Lanqian Feng carried out the experiments. Zhuoxin Chen produced and processed the digital elevation model of erosion landform. Mingming Guo and Wennlong Wang written and prepared the manuscript with contributions from all co-authors.

## Competing interests:

The authors declare that they have no conflict of interest.

## Acknowledgments

This work was supported by the National Natural Science Foundation of China (41571275) and the National Key Research and Development Program of China (2016YFC0501604). Acknowledgement for the data support from "Loess Plateau Data Center, National Earth System Science Data Sharing Infrastructure, National Science & Technology Infrastructure of China. (http://loess.geodata.cn)".

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
