# Peer review of "Spatial-temporal changes in flow hydraulic characteristics and soil loss 1 during gully headcut erosion under controlled conditions 2 3 Mingming Guoa, Zhuoxin Chenb, Wenlong Wangb,c\*, Tianchao Wangd, Qianhua Shib, Hongliang 4 Kangb, Man Zhaob, Langian Fengc 5 6 7 a Key laboratory of Mollisols Agroecology, Northeast Institute of Geography and Agroecology, 8 Chinese Academy of Sciences, Harbin 1500"

_Hydrology and Earth System Sciences, 2020_

## Referee Comment (RC1) · Anonymous Referee #1 · 29 Jan 2021

The manuscript entitled "Spatial-temporal changes in flow hydraulic characteristics 1 and soil loss during gully headcut erosion" is a study of gully head evolution under controlled conditions belonging to a set of papers presented by the authors in the last years on this subject. Since the authors aim in this manuscript is to investigate the hydraulics changes in the flow through the gully and its impact on energy consumption it fits into the journal scope. My general comments on the manuscript are that it covers a topic of interest, and it is well structured with clear Tables and Figures. Not been a native English speaker I find hard to suggest specific changes in the English usage, but there are sections that are hard to follow and expressions that does not seem the most appropriate (e.g. line 146 "artificially planted forest..."). In my opinion the most

valuable part of the manuscript is the experimental dataset presented by the authors, which are scarce in gully erosion studies. However, the results are mostly a confirmation of the previous knowledge, some hardly novel like the transition in hydraulic regime trough the gully headcut, which limits the significance of the manuscript. This might be compensated with a more critical discussion of the results, which currently is mostly a comparison of previous papers. The analysis and discussion of the issue of energy dissipation (consumption in the authors wording), which might be the most innovative part of the manuscript is not deep enough, and does not try to link with previous studies on optimization of energy dissipation in drainage networks which might be enlightening.

Some specific comments that might be of help to the authors for improving the manuscript are:

1- Better description in material and methods of the flume conditions in the gully bed section. It is apparent in Figure 3e that this is a short section with lateral walls, and so it is a situation far off from which might appear on gully in real world conditions, where gully walls will expand at a different rate and energy dissipation might take place for a longer section. Although this does not invalidate the experiment performed by the authors, it clearly conditions the expected results and the conditions to which we could extrapolate the results. This should be considered in the discussion. 2- The discussion of results seems to be mostly a comparison of results for previous papers, with little additional insight. A deeper discussion, which might include, for instance, implications for modelling gully erosion, scale effects (for larger or smaller gullies), or restoration efforts might be included. 3- The analysis if energy dissipation does not seem in depth enough. Firstly, there is no any attempt to provide an overall energy analysis of the system, there are energy losses in the water depth which are not mentioned (like the one dissipated as heat and noise) and it is not clear how much of the original energy available to the flow (which I guess is the potential energy at the reservoir located at the top of the upstream section) is dissipated and how much remains at the end of the flume. The authors do not try to analyze the results to seek if some kind of optimization

of energy dissipation (as suggested by previous papers on river and rill network development) is apparent. For instance calculating the energy losses by wetted section or by unit flow or sediment discharge, like previous studies. These are ideas, among others in that line, that the authors may want to explore to take advantage of the very detailed dataset that they have developed.

4- Data availability. Although in this case it is beyond the authors' responsibility to decide on data availability, it is a good practice to provide at least the ancillary data from which the graphs have been developed. For this I mean the values of the data plotted in the graphs. In this way you facilitate use to other colleagues of the information that might be retrieved scanning and plotting the graphs. I recommend the authors to seek permission for this.

Some other minor technical corrections that I might add are:

1- Title. Perhaps "Spatial-temporal changes in flow hydraulic characteristics and soil loss during gully headcut erosion under controlled conditions" might be more descriptive of the manuscript.

2- Lines 62-64. Not very clear, please edit for clarity.

3- Lines 85-86. Reduces soil losses as and headcut retreat as compare to what, bare soil? Please clarify 4- Lines 133-134. You probably do not need two references to indicate elevation range in the area. 5- Check English usage in line 145-146. 6- Line 215. Please provide a bit more information on the LS300-A measuring principle. 7-section 2.2.2.. Could you indicate the corresponding upslope area for the different flow discharge used, according to the runoff coefficient, and storm intensity used? 8- Lines 241-241. Error in X-Y dimension or in Z dimension? Please clarify. 9- Discussion on soil losses. A mention of the sediment concentration measured in the upstream and flume outlet might be quite helpful to understand the erosion/deposition processes.

412, 2020.

---

## Referee Comment (RC2) · Artemi Cerdà (Referee) · 22 Feb 2021

Dear author I found your paper of interest and a good piece of science I suggest some updates that are necessary in my opinion Please, check the comments on the figures layout You can improve and make the figures more relevant The introduction and discussion sections are based on old and european researches, and in the last five years scientists from Iran, China and Ethiopia developed new research on gully geomorphology, hydrology and spatial distribution I recommend to update your literature review Sincerely Artemi Cerdà

[Figure]

Please also note the supplement to this comment:
https://hess.copernicus.org/preprints/hess-2020-412/hess-2020-412-RC2-supplement.pdf

**Supplement:**

[revised manuscript text omitted]

---

## Author Response (AR1)

**Reply to Editor**

**Wenlong Wang on behalf of all co-authors**

Editor Decision: Publish subject to revisions (further review by editor and referees) (27 Mar 2021) by Thom Bogaard

Comments to the Author:

Dear authors

You received two detailed and constructive reviews and I really liked your reply to them. Your work fits in HESS and has some innovative aspects, mainly related to the energy dissipation and soil loss in headcuts. Also, the agreed deepening of the discussion of implication of your findings are welcomed and will improve the impact of your work. I was pleased to read you will make all data of your interesting experiments available. This can be done attached to the paper. I look forward to the revised version which will go for final review by one reviewer and myself.

Best Thom Bogaard

**Response:** Thank you for your letter and for the decision concerning our manuscript entitled "Spatial-temporal changes in flow hydraulic characteristics and soil loss during gully headcut erosion" (ID: hess-2020-412). First, we really thank you for your recognition about some innovative aspects of our work (e.g., energy dissipation and soil loss in headcuts). These comments are valuable and very helpful for revising and improving our paper, as well as the important guiding significance to our further researches. At the same time, we thank you for giving us the opportunity to revise the manuscript. We have studied and analyzed comments carefully and have made many corrections which we hope meet with approval. To clearly respond all comments, point by point, these comments from reviewers were classified by authors based on the specific meanings and listed as following (Q1, Q2, Q3…). The main revisions in the paper and the responds to the reviewer's comments are as following.

**Reply to Referee #1**

**Wenlong Wang on behalf of all co-authors**

The manuscript entitled "Spatial-temporal changes in flow hydraulic characteristics and soil loss during gully headcut erosion" is a study of gully head evolution under controlled conditions belonging to a set of papers presented by the authors in the last years on this subject. Since the authors aim in this manuscript is to investigate the hydraulics changes in the flow through the gully and its impact on energy consumption it fits into the journal scope. My general comments on the manuscript are that it covers a topic of interest, and it is well structured with clear Tables and Figures. Not been a native English speaker I find hard to suggest specific changes in the English usage, but there are sections that are hard to follow and expressions that does not seem the most appropriate (e.g. line 146 "artificially planted forest: : :"). In my opinion the most valuable part of the manuscript is the experimental dataset presented by the authors, which are scarce in gully erosion studies. However, the results are mostly a confirmation of the previous knowledge, some hardly novel like the transition in hydraulic regime trough the gully headcut, which limits the significance of the manuscript. This might be compensated with a more critical discussion of the results, which currently is mostly a comparison of previous papers. The analysis and discussion of the issue of energy dissipation (consumption in the authors wording), which might be the most innovative part of the manuscript is not deep enough, and does not try to link with previous studies on optimization of energy dissipation in drainage networks which might be enlightening.

**Response:** First of all, we would like to thank you for your recognition of our work that is a topic of interest and fits within the scope of the journal. As for the language, we will revise it carefully several times, and then invite researchers who have been working in Europe for a long time to revise the manuscript, hoping to make the language smoother. Also, we thank you for your recognition of our research work on gully erosion and the value of the data. We also agree your comments about "the results are mostly a confirmation of the previous knowledge, some hardly novel like the transition in hydraulic regime trough the gully headcut, which limits the significance of the manuscript. This might be compensated with a more critical discussion of the results, which currently is mostly a comparison of previous papers.". We will reduce the comparison with previous

studies and compensate the discussion in more depth about what our results show and why this is the case than the simple comparison with previous studies. One of the important reasons why the process of gully erosion is more complicated than that of slope erosion is that the existence of gully head changes the runoff characteristics and erosion dynamic mechanism of slope. For a complete gully system, the flow characteristics of upstream area, gully head runoff and gully bed are completely different, and the erosion process and hydro-dynamic mechanism of the corresponding three landform parts are also completely different. Therefore, this is also the reason why we study the change of flow properties along upstream area - gully head - gully bed and its influences on runoff energy consumption and soil loss in the process of gully headcut erosion. We believe that understanding the erosion process and hydrodynamic mechanism of different landforms units is conducive to a deeper understanding of the process and mechanism of gully erosion, and provides some references for the development and establishment of gully erosion process models.

In addition, as you mentioned, the energy dissipation may be the most innovative part of the manuscript. We also found the soil loss due to gully headcut erosion had closer correlation with energy dissipation than other hydraulic parameters. So, we will intensify this in-depth analysis and discuss it with previous studies on optimization of energy dissipation in drainage networks. The more detailed explanation is also provided in Q3 and specific revision was reflected in the revised manuscript. We are particularly grateful to you for your valuable suggestions, which will make our work more in-depth and excellent.

**Some specific comments that might be of help to the authors for improving the manuscript are:**

**Q1.** Better description in material and methods of the flume conditions in the gully bed section. It is apparent in Figure 3e that this is a short section with lateral walls, and so it is a situation far off from which might appear on gully in real world conditions, where gully walls will expand at a different rate and energy dissipation might take place for a longer section. Although this does not invalidate the experiment performed by the authors, it clearly conditions the expected results and the conditions to which we could extrapolate the results. This should be considered in the discussion.

**Response:** First of all, we are very grateful for your valuable suggestions for amendments. (1) We have added more descriptions about the gully bed in the "2.2.1 Plot set-up" in "2 materials and methods" section. The specific revision is "According to the pre-experimental results, the length of upstream area can meet the needs of headcut migration under designed flow discharge $(3.0 - 7.2 \text{ m}^3$

h$^{-1}$) and gully head height (0.9 m), and the length of gully bed also can satisfy the development of plunge pool by jet flow and stabilize the flow of gully bed." (2) Gully erosion has three sub-processes, including gully headcut erosion, gully bank expansion and gully incision. Our research mainly focuses on gully headcut erosion, which is consistent with the research of many people such as Bennet et al. (1999, 2000, 2001, 2006), Gordon et al. (2007), Wells et al. (2009a, 2009b), Su et al. (2013, 2014), Zhang et al. (2016, 2018) and Guo et al. (2019, 2021). So, in this study only when the gully head migrates upstream, the gully channel and gully bank will be formed, so the gully bed section designed in this experiment does not include the gully bank. In this study, the gully bed was set up for the development of plunge pool, and also for clarifying the changes in the runoff characteristics of the upstream area and the gully bed and their impact on the erosion dynamic mechanism due to the existence of the gully heads. We also did some pre-experiments and found that the length of the gully bed can meet the morphological development of the plunge pool, and it can also stabilize the water flow after it comes out of the plunge pool (because the most of flow energy was consumed in plunge pool, so the flow energy and velocity are very low and easy to stable). As you stated, indeed, when gully head migrates, the flow energy dissipation might take place for a longer section than the designed gully bed section. In our study, we treated this as the result of gully head migration, and thus the energy dissipation due to gully walls expand was included in the range of gully head energy consumption. Of course, to some extent, this does clearly condition the expected results and the conditions to which we could extrapolate the results. We have added the above related content to the Discussion section according to your suggestion. Thanks again.

● Bennett, S.J., Casali, J.: Effect of initial step height on headcut development in upland concentrated flows. Water Resources Research, 37, 1475–1484, https://doi.org/10.1029/2000WR900373, 2001.
● Bennett, S.J.: Effect of slope on the growth and migration of headcuts in rills, Geomorphology, 30, 273–290, https://doi.org/10.1016/S0169-555X(99)00035-5, 1999.
● Bennett, S.J., Alonso, C.V.: Turbulent flow and bed pressure within headcut scour holes due to plane reattached jets, Journal of Hydraulic Research, 44, 510–521, https://doi.org/10.1080/00221686.2006.9521702, 2006.
● Bennett, S.J., Alonso, C.V., Prasad, S.N., Romkens, M.J.: Experiments on headcut growth and migration in concentrated flows typical of upland areas, Water Resources Research, 36, 1911–1922, https://doi.org/10.1029/2000WR900067, 2000.
● Su, Z.A., Xiong, D.H., Dong, Y.F., Zhang, B.J., Zhang, S., Zheng, X.Y., …Fang, H.D.: Hydraulic properties of concentrated flow of a bank gully in the dry-hot valley region of southwest

China, Earth Surface Processes and Landforms, 40, 1351–1363. https://doi.org/10.1002/esp.3724, 2015.

● Su, Z.A., Xiong, D.H., Dong, Y.F., Li, J.J., Yang, D., Zhang, J.H., He, G.X.: Simulated headward erosion of bank gullies in the Dry-hot Valley Region of southwest China, Geomorphology, 204, 532–541, https://doi.org/10.1016/j.geomorph.2013.08.033, 2014.

● Wells, R.R., Alonso, C.V., Bennett, S.J.: Morphodynamics of Headcut Development and Soil Erosion in Upland Concentrated Flows, Soil Science Society of America Journal, 73, 521–530. https://doi.org/10.2136/sssaj2008.0007, 2009a.

● Wells, R.R., Bennett, S.J., Alonso, C.V.: Effect of soil texture, tailwater height, and pore-water pressure on the morphodynamics of migrating headcuts in upland concentrated flows, Earth Surface Processes and Landforms, 34, 1867–1877, https://doi.org/10.1002/esp.1871, 2009b.

● Zhang, B.J., Xiong, D.H., Su, Z.A., Yang, D., Dong, Y.F., Xiao, L., Zhang, S., Shi, L.T.: Effects of initial step height on the headcut erosion of bank gullies: a case study using a 3D photo-reconstruction method in the Dry-hot Valley region of southwest China, Physical Geography, 37, 409–429, https://doi.org/10.1080/02723646.2016.1219939, 2016.

● Zhang, B.J., Xiong, D.H., Zhang G.H., Zhang, S., Wu, H., Yang, D., Xiao, L., Dong, Y.F., Su, Z.A., Lu, X.N.: Impacts of headcut height on flow energy, sediment yield and surface landform during bank gully erosion processes in the Yuanmou Dry-hot Valley region, southwest China, Earth Surface Processes & Landforms, 43(10), 2271-2282, https://doi.org/10.1002/esp.4388, 2018.

● Guo, M., Wang, W., Shi, Q., Chen, T., Kang, H., Li, J.: An experimental study on the effects of grass root density on gully headcut erosion in the gully region of China's Loess Plateau, Land Degradation & Development, 30, 2107–2125, https://doi.org/10.1002/ldr.3404, 2019.

● Guo, M.M., Lou, Y.B., Chen, Z.X., Wang, W.L., Feng, L.Q., Zhang, X.Y.: The proportion of jet flow and on-wall flow and its effects on soil loss and plunge pool morphology during gully headcut erosion, Journal of Hydrology, 598, 126220, https://doi.org/10.1016/j.jhydrol.2021.126220, 2021.

**Q2.** 2- The discussion of results seems to be mostly a comparison of results for previous papers, with little additional insight. A deeper discussion, which might include, for instance, implications for modelling gully erosion, scale effects (for larger or smaller gullies), or restoration efforts might be included.

**Response:** We thank you for your useful comments and valuable suggestions regarding the discussion. We have also realized the shortcomings of the discussion part, and we have reduced some comparison with previous studies, and also supplemented and improved each discussion part to increase the intrinsic significance of our research results. We added some aspects about the explanation about how the hydraulic characteristics of three landform units with gully headcut, the effect of gully headcut on variation of hydraulic properties from upstream to gully bed, and clarify the difference in hydrodynamic mechanism of three landform units during headcut erosion. In addition, we also added a section (**5 Implication, significance and limitations of this study**) focusing on the implications and significance of our research about gully erosion for improving the

significance and value of this study. This section referred to some points including modeling gully erosion, scale effect of gully erosion, establishment of gully erosion model and gully erosion control practice, etc.

**Q3.** 3- The analysis if energy dissipation does not seem in depth enough. Firstly, there is no any attempt to provide an overall energy analysis of the system, there are energy losses in the water depth which are not mentioned (like the one dissipated as heat and noise) and it is not clear how much of the original energy available to the flow (which I guess is the potential energy at the reservoir located at the top of the upstream section) is dissipated and how much remains at the end of the flume. The authors do not try to analyze the results to seek if some kind of optimization of energy dissipation (as suggested by previous papers on river and rill network development) is apparent. For instance calculating the energy losses by wetted section or by unit flow or sediment discharge, like previous studies. These are ideas, among others in that line, that the authors may want to explore to take advantage of the very detailed dataset that they have developed.

**Response:** We fully agree with your comment. Indeed, we have not provided the overall energy analysis of the system in the original submission. As you guessed, the original energy available to flow is the potential energy based on the zero-potential surface (the bottom of the gully bed section), which was calculated by Eq. (9). We will add the analysis about the "how much of the original energy available to the flow is dissipated and how much remains at the end of the flume". The following figure shows the total flow energy, total energy consumption and the rest flow energy under different flow discharge conditions. The added Fig. 1 will combinate with Figure 10 in "3.2 Spatial-temporal change of energy consumption".

In addition, we indeed have not analyzed the results to seek if some kind of optimization of energy dissipation (as suggested by previous papers on river and rill network development).

Previous studies showed that the similarity between rill/gully networks and river drainage networks is not surprising since rills belong to the same drainage network that river sections do, and some lab studies have documented similarity between river and rill network (e.g., Mosley, 1974; Ogunlela, et al., 1989) as well as the similarity between drainage networks at basin and small scales (Helming et al., 2006). That is why that, in the soil erosion field, the flow dynamic knowledges/theory of river systems are always used to understand and model rill/gully network or hillslope erosion processes.

At present, the theory of minimum rate of energy dissipation have been used to describe channels in equilibrium with water and sediment based on analogy with thermodynamic systems (Yang, 1971a, 1971b; Yang et al., 1981) and also was used in the studies on the hydro-dynamic mechanism of soil erosion due to above-mentioned similarity. The theory of minimum rate of energy dissipation was expressed as following equations (Yang, 1971a, 1971b) and also employed in our study.

The flow energy and unit flow power are:

$E=\rho \cdot g \cdot l \cdot Q \cdot J$

$P=V \cdot J$

Where $\rho$ is water density, $l$ is channel length, $Q$ is the unit flow discharge, $V$ is flow velocity, $J$ is the slope gradient.

In our study, the calculation of flow energy in different landform units (Eqs. (9-12)) is consistent with the two principles.

In addition, we also analyzed the temporal change in the ratio of total energy dissipation to total flow energy with experimental time (Figure 2 as following) and find the similar trend with the published article by Gomez et al. (2003) who found that the theory developed for river networks might explain the evolution of rill networks at hillslope scale. Therefore, we also further confirmed that the flow energy dissipation in our study followed the theory of minimum rate of energy dissipation.

To be honest, nevertheless, many studies distinguished between rill/gully and river scales because there are significant differences between them due to their different sizes and the more discontinuous and ephemeral character of the rill/gully compared with rivers. This difference in size implies that some mechanisms involve in the evolution of rill/gully networks may seldom be present in the evolution of river networks. For that reasons, the interpretation of rill/gully networks process form the experimental and theoretical studies of river networks may not be straightforward, in same way that the interpretation of the evolution of river networks form small-scale rill studies has its shortcomings. Throughout the current researches on the hydrodynamic mechanism of soil erosion, most of the hydrodynamic calculations are carried out according to the river dynamics formula, and in terms of energy consumption, the principle of minimum energy consumption is used.

We will analyze the results to clarify which kind of optimization of energy dissipation is available and build the connection between soil loss and energy dissipation for the "UA-GH-GB" gully system

in our study and promise to revise the related content in revised MS (Fig. 3). The specific addition could be check in Figure 9 and Figure 13 of revised MS.

Also, our research team will do our best to study the hydrodynamic mechanism of gully erosion to replace the theory of river dynamics.

References:

Mosley, M.P., 1974. Experimental study of rill erosion, Trans. ASAE, 17(5), 909-916.

Ogunlela, A., Wilson, B.N., Rice, C.T., Couger, G., 1989. Rill network development and analysis under simulated rainfall, Pap. 89-2112, Am. Soc. of Agric. Eng., St. Joseph, Mich.

Helming K , Rmkens M J M , Prasad S N , et al. Erosional development of small scale drainage networks[M]. Springer Berlin Heidelberg, 2006.

Yang, C.T., 1971a. Potential energy and stream morphology, Water Resource Research, 7(2), 311-223.

Yang, C.T., 1971b. On river meanders, Journal of Hydrology, 13, 231-253.

Yang, C.T., Song, C.S.S., Woldenberg, M.J., 1981. Hydraulic geometry and minimum rate of energy dissipation, Water Resource Research, 17(4), 1014-1018.

Gómez, J.A., Darboux, F., Nearing, M.A., 2003. Development and evolution of rill networks under simulated rainfall. Water Resources Research, 39(6), 1148.

[Figure]

Fig. 1 Variations in total energy consumption and rest flow energy with flow discharge and their ratios to total energy

[Figure]

Figure 2 The ratio of total energy consumption to total flow energy

[Figure]

Fig. 3 Relationship between soil loss rate of "UA-GH-GB" system and energy dissipation during headcut erosion

**Q4.** 4- Data availability. Although in this case it is beyond the authors responsibility to decide on data availability, it is a good practice to provide at least the ancillary data from which the graphs have been developed. For this I mean the values of the data plotted in the graphs. In this way you facilitate use to other colleagues of the information that might be retrieved scanning and plotting the graphs. I recommend the authors to seek permission for this.

**Response:** We fully understand your comment. As you mentioned in general comments, "In my

opinion the most valuable part of the manuscript is the experimental dataset presented by the authors, which are scarce in gully erosion studies.". At present, indeed, the datasets about gully erosion process are scarce. After a careful consideration, we obtain the permission and can provide the data information of all figures in the manuscript. In the revision system, we have provided and uploaded the data plotted in the figures for other colleagues at the separated file in the submission system. The data file is named as **Figure's data (.xls)**.

**Some other minor technical corrections that I might add are:**

**Q5.** 1- Title. Perhaps "Spatial-temporal changes in flow hydraulic characteristics and soil loss during gully headcut erosion under controlled conditions" might be more descriptive of the manuscript.

**Response:** We thank you for your pertinent suggestion. Indeed, our study was designed and completed under controlled conditions. We accepted your suggestion and will revised the title as "Spatial-temporal changes in flow hydraulic characteristics and soil loss during gully headcut erosion under controlled conditions". Thank you again.

**Q6.** 2- Lines 62-64. Not very clear, please edit for clarity.

**Response:** We thank you for your careful review. We will revise the sentence in future revised MS as "Moreover, the different landform units (upstream area, UA; gully head, GH; gully bed, GB) of gully system exhibited completely different erosion processes and hydrodynamic mechanisms during gully headcut erosion (Zhang et al., 2018; Guo et al., 2019; Shi et al., 2020a).". We hope our amendments will be satisfactory to you.

**Q7.** 3- Lines 85-86. Reduces soil losses as and headcut retreat as compare to what, bare soil? Please clarify

**Response:** We are so sorry for the unclear expression. As you said, the reduced soil losses were compared with bare soil. We will revise the sentence as "Guo et al. (2019) concluded that the grass (Agropyron cristatum) could reduce soil loss and headcut retreat distance by 45.6–68.5%, 66.9–85.4%, respectively, compared with bare soil, and the roots of 0–0.5 mm in diameter showed the greatest controlling influence on headcut erosion." in the future revision. Thank you very much for your careful review.

**Q8.** 4- Lines 133-134. You probably do not need two references to indicate elevation range in the area.

**Response:** We are sorry for it. In fact, we should insert the two references in the last sentence. We will revise the sentence as "The mean annual precipitation is 546.8 mm (1954 - 2014), of which precipitation from May to September accounts for 76.9% of the total precipitation (Xia et al., 2017; Guo et al., 2019). The elevation ranges from 1050 to 1423 m.". Thank you very much.

**Q9.** 5- Check English usage in line 145-146.

**Response:** We thank you for your careful review. We will revise the sentence as "The plants are primarily artificially planted arbors and herbaceous vegetation and shrubs" in the future revision.

**Q10.** 6- Line 215. Please provide a bit more information on the LS300-A measuring principle.

**Response:** We thank you for your valuable suggestion. We will add the more information on the measuring principle. The specific revision is "The runoff velocity ($V_J$) before runoff arrived at the brink of headcut was measured 5 – 8 times by the flow velocity measuring instrument (LS300-A). The instrument was firstly placed perpendicular to the flow section but does not touch the underlying surface. When the flow passes through the turbine, the flow velocity can be measured by the rotating velocity of the turbine with the accuracy of 0.01 m s$^{-1}$ and measuring error of < 1.5%, Also, the runoff width at the headcut brinkpoint was measured (Fig. 3d).".

**Q11.** 7- section 2.2.2. Could you indicate the corresponding upslope area for the different flow discharge used, according to the runoff coefficient, and storm intensity used?

**Response:** We thank you for your meaningful comment. In fact, the five different flow discharges were selected from a range values (3.12 to 9.68 m$^3$ h$^{-1}$) that was calculated according to the runoff coefficient, storm intensity and upstream area. The runoff coefficient was confirmed based on the data of standard runoff plots; the storm intensity was calculated by Eq. (1). For the upstream area, our research team investigated 45 upstream areas in the study area. The upstream area (A) and width (W) are 0.15-8.69 km$^2$ and 0.53-1.64 km, respectively. **Then, the inflow discharge was calculated by Eq. (2) and ranged from 3.12 to 9.68 m$^3$ h$^{-1}$. Before the study, we first conducted some**

**preliminary experiments under some flow discharges, and meanwhile considering the pre-experiment effect, finally, we selected the five inflow discharge levels (3.0, 3.6, 4.8, 6.0, and 7.2 m³ h⁻¹) from the calculated ranges (3.12 to 9.68 m³ h⁻¹). Therefore, the five discharges had not the investigated corresponding upslope area in the study, but the five flow discharges are bound to happen in the real situation. In fact, the same upslope drainage area may correspond different unit width flow discharges due to the different length or width of catchment area.**

$$RI = \frac{5.09N^{0.379}}{(t+1.4)^{0.74}} \quad (1)$$

where $RI$ is the average rainfall intensity during $t$ minutes, mm min⁻¹; $N$ is the recurrence period of rainstorm, yr; and $t$ is the rainfall duration, min.

$$q = \frac{60\alpha \cdot A \cdot RI \cdot w}{W} \quad (2)$$

where $w$ is the plot width, m; and $\alpha$ is the runoff coefficient of bare land and is identified as 0.167 by analyzing the runoff and rainfall data of standard runoff plots (Li et al., 2006).

**Q12.** 8- Lines 241-241. Error in X-Y dimension or in Z dimension? Please clarify.

**Response:** We are so sorry for the unclear expression. We will revise the sentence as "The root mean square errors for the altitudes (Z axis) of the target points are 0.0037, 0.0045, 0.0024, 0.0052 and 0.0030 m on average, respectively, for the experiments of five inflow discharges, which can satisfy the study requirement (millimeter level)".

**Q13.** 9- Discussion on soil losses. A mention of the sediment concentration measured in the upstream and flume outlet might be quite helpful to understand the erosion/deposition processes.

**Response:** We are particularly grateful to you for your valuable suggestions on the revision of our manuscript. As you mentioned, we indeed measured/calculated the sediment concentration of upstream area and flume outlet. The sediment concentration of upstream area and flume outlet was showed as following figures (Fig. 4) under different flow discharge conditions. In fact, the sediment concentration of flume outlet can reflect the erosion process of the "upstream area - gully head – gully bed" system. Compared with the soil loss process of the upstream area and the system (Figure 11a, 11b), we found the change in sediment concentration with time is consistent with the temporal change of soil loss rate. Moreover, in Figure 11, we exhibited the soil loss/deposition process of

three landform units as well as the "upstream area - gully head – gully bed" system, which can reflect the erosion/deposition process of each landform unit.

We also think the mention of sediment concentration is worth adding. However, given this similar change trend between soil loss process and sediment concentration process, we carefully considered and decided not to add it in the results section. Of course, We respect the suggestions of reviewers very much, and thus, we have added some related discussion in Discussion section about soil losses.

[Figure]

Fig. 4 Temporal change in sediment concentration of upstream area and flume outlet

[Figure]

Figure 11 (original MS)

**Reply to Referee #2**

**Wenlong Wang on behalf of all co-authors**

Dear author I found your paper of interest and a good piece of science. I suggest some updates that are necessary in my opinion. Please, check the comments on the figure's layout. You can improve and make the figures more relevant. The introduction and discussion sections are based on old and European researches, and in the last five years scientists from Iran, China and Ethiopia developed new research on gully geomorphology, hydrology and spatial distribution I recommend to update your literature review.

**Response:** We particularly appreciate your recognition of our study. We have checked all the figures and updated them according to your opinions. The specific modifications can be checked in **Q1-Q11**, and we have also modified and updated them in the revised MS. In addition, in the introduction and discussion section, we will add the literatures involved gully erosion, gully geomorphology, hydrology and spatial distribution in the last five years (published in China, Iran, Ethiopia and other countries) in the revised manuscript. Thank you very much again.

**The following questions was list according to the PDF file named by "hess-2020-412-RC2-supplement" provided by Reviewer 2#.**

**Q1. L50**-The literature used in the introduction is the right one but is mainly focuses in an West European perspective. There are researchers in Ethiopia, China and Iran that recently published relevant papers that should be mentioned. In general, the literature review is too old.

> 49   erosion accounts for 10% - 94% of total soil loss amount based on the collected data from published
>
> 50   articles. Moreover, gully erosion can severely damage to infrastructure, enhance the terrain
>
> 51   fragmentation, and cause ecosystem instability, land degradation and food safety (Poesen et al., 2003;
>
> 52   de Vente & Poesen, 2005; Li et al., 2015; Vanmaercke et al, 2016; Hosseinalizadeh et al., 2019).
>
> 53         As one of the gully erosion processes, the gully headcut retreat often significantly influences

> The literature used in the introduction is the right one but is mainly focuses in an West European perspective. There are researchers in Ethiopia, China and Iran that recently published relevant papers that should be mentioned. In general the literature review is too

**Response**: We thank you for your valuable suggestion. Indeed, the several literatures in L50-53 are old. We research and select the following literatures according to the information you provided. As a result, the sentence was revised as "Moreover, gully erosion can severely damage to infrastructure,

enhance the terrain fragmentation, and cause ecosystem instability, land degradation and food safety (Vanmaercke et al., 2016; Zhang et al., 2018; Hosseinalizadeh et al., 2019; Arabameri et al., 2020; Bogale et al., 2020; Belayneh et al., 2020; Wen et al., 2020).". The references were listed as following:

Zhang, X., Fan, J., Liu, Q., Xiong D.: The contribution of gully erosion to total sediment production in a small watershed in Southwest China, Physical Geography, 39(3), 1-18, https://doi.org/10.1080/02723646.2017.1356114, 2018.

Arabameri, A., Chen, W., Lombardo, L., Blaschke, T., Tien Bui, D.: Hybrid computational intelligence models for improvement gully erosion assessment, Remote Sensing, 12(12), https://doi.org/10.3390/rs12010140, 140, 2020.

Bogale, A. G., Aynalem, D. W., Adem, A. A., Mekuria, W., Tilahun, S.: Spatial and temporal variability of soil loss in gully erosion in upper Blue Nile basin, Ethiopia, Applied Water Science, 10(5), 106, https://doi.org/10.1007/s13201-020-01193-4, 2020.

Belayneh, M., Yirgu, T., Tsegaye, D.: Current extent, temporal trends, and rates of gully erosion in the Gumara watershed, northwestern Ethiopia, Global Ecology and Conservation, 24, e01255, https://doi.org/10.1016/j.gecco.2020.e01255, 2020.

Wen, Y., Kasielke, T., Li, H., Zhang, B., Zepp, H.: May agricultural terraces induce gully erosion? a case study from the black soil region of northeast China. Science of The Total Environment, 750(4), 141715, https://doi.org/ 10.1016/j.scitotenv.2020.141715, 2020.

**Q2. L-136** I miss here information about the land use now and the past land uses

**Response:** The main land use on loess-tableland position has always been farmland and orchards, while the land use on hillslope is sloping farmland and orchards before 1999, which have been changed into forested and grassy land due to the Chinese Grain for Green program. These results have been added in the revised manuscript.

**Q3. L-145** I suggest to use Mg instead of t

**Response:** We thank you for your suggestion, and we have revised the "4350 t km$^{-2}$ y$^{-1}$" as "4350 Mg km$^{-2}$ y$^{-1}$" in the Revised MS.

**Q4. L-174**

173    recurrence period of "A" type rainstorm was designed as 30 years. Previous studies indicated that the

174    rainstorm distribution on the Loess Plateau showed a non-significant change  in past decades (Li et

**Response:** We thank you for your careful review. We have deleted the space in revised MS. The sentence will be revised as "Previous studies indicated that the rainstorm distribution on the Loess Plateau showed a non-significant change in past decades (Li et al., 2010; Sun et al., 2016; Wen et al., 2017)."

**Q5. L-554** your discussion section needs an update on the literature and the topics. see here some recent papers that can help.

553    describing soil detachment process. This comparison also fully illustrated the great difference in

554    hydrodynamic characteristic between hillslope erosion and headcut erosion. In this study, the soil

555    loss of gully head (including plunge pool erosion) was significantly affected by jet properties. It's

556    confirmed that the plunge pool erosion by jet flow becomes a crucial process controlling gully head

557    migration and sediment production (Oostwoud-Wijdenes et al., 2000). Consequently, the plunge pool

558    erosion theory is usually employed to build several headcut retreat models (Alonso et al., 2002;

559    Campo-Bescós et al., 2013). Although the weak correlation between soil loss of gully head and flow

560    velocity at headcut breakpoint, the larger flow velocity resulted from increasing inflow discharge

561    would improve the shear stress of jet flow impinging gully bed, and thus the gully headcut suffered

your discussion section needs an update on the literaure and the topics
see here some recent papers that can help

Li, Y., Mo, Y. Q., Are, K. S., Huang, Z., Guo, H., Tang, C., ... & Wang, X. (2021). Sugarcane planting patterns control ephemeral gully erosion and associated nutrient losses: Evidence from hillslope observation. *Agriculture, Ecosystems & Environment*, *309*, 107289.

Amare, S., Keesstra, S., van der Ploeg, M., Langendoen, E., Steenhuis, T., & Tilahun, S. (2019). Causes and controlling factors of Valley bottom Gullies. *Land*, *8*(9), 141.

Sidorchuk, A. (2020). The potential of gully erosion on the Yamal peninsula, West Siberia. *Sustainability*, *12*(1), 260.

Amare, S., Langendoen, E., Keesstra, S., Ploeg, M. V. D., Gelagay, H., Lemma, H., & van der Zee, S. E. (2021). Susceptibility to Gully Erosion: Applying Random Forest (RF) and Frequency Ratio (FR) Approaches to a Small Catchment in Ethiopia. *Water*, *13*(2), 216.

**Response:** We are particularly grateful to you for providing us with new literature. We have added them in the revised version. Since this manuscript was submitted in July 2020, some of the latest literature may not have been retrieved and cited. In order to make the discussion more complete, we have also searched more literatures related to this study, and will supplement them in the revised MS.

Thank you again.

**Q6. L-848—Figure 1**-Your research is in a laboratory in the field (controlled conditions). You do not need this map or some information of the study area. I suggest to remove the frames and the 0"

[Figure]

848

849 Figure 1 The location of the experimental site in Nanxiaohegou watershed, Qingyang City, Loess
850 Plateau, China. Note: The figure production was based on the digital elevation model data (spatial
851 resolution of 30 m) which is available from http://srtm.csi.cgiar.org (Reuter et al., 2007).

ⅱ     2021/02/22 22:37:43 ✕

Your research is in a laboratory in the field (controlled conditions)
You do not need this map or some information of the study area

ⅱ     2021/02/22 22:35:28 ✕

I suggest to remove the frames and the 0''

**Response:** We thank you for your valuable suggestion. After a careful consideration, indeed, our study is in a lab in the field under controlled conditions. The information of study area has been described in Materials and Method section. We decided to take your advice and have deleted the Figure 1. Thank you very much.

**Q7. L-863—Figure 5**

[Figure]

863
864            Fig.5 Temporal changes in jet properties of headcut and their relationships with inflow discharge.
865

**Response: We thank you for your valuable suggestion. The Figure 5 have been revised as following:**

[Figure]

**Figure 5** Temporal changes in jet properties of headcut and their relationships with inflow discharge.

**Q8. L-866—FIGURE 6**

866

[Figure]

867
868       Fig.6 Temporal changes in runoff regime of upstream area and gully bed and their relationships with inflow
869                                    discharge.

**Response: We thank you for your valuable suggestion. The Figure 6 have been revised as following:**

[Figure]

**Figure 6** Temporal changes in runoff regime of upstream area and gully bed and their relationships with inflow discharge.

**Q9. L-872—FIGURE 8**

[Figure]

 Fig.8 Temporal changes in runoff shear stress and stream power of upstream area and gully bed and their
 relationships with inflow discharge

**Response: We thank you for your valuable suggestion. The Figure 8 have been revised as following:**

[Figure]

**Q10.    L876—Figure 9**

[Figure]

876
877     Fig.9 Temporal changes in runoff energy consumption of upstream area, gully head and gully bed under different
878                     inflow discharge conditions
879

**Response: We thank you for your valuable suggestion. The Figure 9 have been revised as following:**

[Figure]

**Figure 9** Temporal changes in runoff energy consumption of upstream area, gully head and gully bed under different inflow discharge conditions

**Q11. L-880—Figure 10**

[Figure]

880
881 Fig.10 The variation in energy consumption of upstream area, gully head and gully bed and their proportions with
882 inflow discharge
883

Response: We thank you for your valuable suggestion. The Figure 10 have been revised as following:

[Figure]

**Figure10** Variation in energy consumption of upstream area, gully head and gully bed and their proportions with inflow discharge

---

## Author Response (AR2)

**Reply to Editor**

**Wenlong Wang on behalf of all co-authors**

Editor Decision: Publish subject to minor revisions (review by editor) (21 Jul 2021) by Thom Bogaard

Comments to the Author:

Dear authors,

I happy to inform you I have accepted your manuscript with minor revisions for publication in Hess. I think the revised version is well improved. I have a few technical remarks that I would like you to incorporate when submitting the final version which relate to summary / conclusion section and the figures

**Response:** Thank you for your letter and for the decision (Publish subject to minor revisions) concerning our manuscript (ID: hess-2020-412). The following comments are valuable and very helpful for improving our work. At the same time, we thank you for giving us the opportunity to revise the manuscript. We have studied and analyzed comments carefully and have made many corrections which we hope meet with approval.

**Q1.** Can you rephrase the summary in terms of conclusions. It is not so useful to repeat the percentages. Please write the last section in terms of scientific conclusions (answering your research objectives.

**Response:** Thank you for your valuable comment. We have revised the Summary section. We deleted some percentages and revised some sentences. The spatial change in energy consumption and soil loss during headcut erosion is the core objectives of this study, and thus some percentages are still retained in Summary. The specific revision could be check in the revised manuscript.

**Q2.** Second, please check English in part Data Availability. I would like to urge you to make the data available as much as possible.

**Response:** Thank you for you suggestion. We are pleasure to share our data for other colleagues. However, after reconfirmation, the original data of this study is not accessible according to the requirement of the funded program. We will share the all data related to all figures in this study, which can fully satisfy the requirement of other colleagues. The data that support the findings of this study are available from the first author (guomingming@iga.ac.cn) and corresponding author upon request (nwafu_wwl@163.com).

Third: Figures.

**Q3.** Fig 2: photo panel description should be in figure caption (Figue 3. Panel a) ..., b) ... etc.

**Q4.** It is OK to indicate things in the photo but the figure description should be below. Please only use black and white for letters, lines and symbols as 'colors and especially red is not visible to everybody. Also try to align the text. Use black or white also for panel indication without grey background

Response: We revised the Figure 2 according to your suggestion. The revised figure 2 as following. Besides, the figure 1 was also revised.

[Figure]

**Figure 2.** Plot construction (a), runoff width measurement of loess-tableland and runoff and sediment sampling of outlet (b), runoff velocity measurement of loess-tableland (c), jet velocity measurement of gully head (d), runoff velocity and width measurement of gully bed (e), and experimental process recoding (f)

[Figure]

**Figure 1.** Sketch (a) and photo (b) of experimental plot

**Q1.** Fig 3: same

Response: The figure 3 was revised as following:

[Figure]

**Figure 3.** Sketch of jet flow at gully headcut (a) and plunge pool at gully bed (b)

**Q2.** Fig 4: in panel b-d-f, please align formula info (put all three in same location in their panel)

**Response: The figure 4 was revised as following:**

[Figure]

**Figure 4.** Temporal changes in jet properties of headcut and their relationships with inflow discharge

**Q3.** Table 1: please add some space between constants and symbols to improve readibility. All formulas are at 0.01 significant so it seems to me the double asterix can go out

**Response: The table 1 was revised as following:**

**Table 1.** The relationships between jet properties of gully headcut and time

[revised manuscript text omitted]

**Q9.** Fig 12: try to align formula text in panels to be at same height and distance from side

**Response: The figure 12 was revised as following:**

[Figure]

**Figure 12.** Relationships between soil loss rate of upstream area, gully bed and gully head and runoff hydraulic and jet properties

**Q10.** Fig 13: use two grades of grey to indicate parts, use black panel indications

**Response: The figure 13 was revised as following:**

[Figure]

**Figure 13.** Synchronous change of soil loss rate of "upstream area-gully head-gully bed" system and total energy dissipation during headcut erosion

**Q11.** Fig 14: black text. Formula does not need bold letter font

**Response: The figure 14 was revised as following:**

[Figure]

**Figure 14.** Relationships between soil loss rate of "upstream area-gully head-gully bed" system (a), upstream area (b), gully head (c) and gully bed(d) and energy consumption

**Q12. Other figures and revisions**

**Response: We thank you for your valuable suggestion. We also revised the Figures 1, 6,8,11.**

**We also check the full MS carefully and revised some inappropriate description.**

---

## Author Response (AR3)

**Reply to Editor**

**Wenlong Wang on behalf of all co-authors**

Editor Decision: Publish subject to technical corrections (23 Jul 2021) by Thom Bogaard

Comments to the Author:

Dear authors

I thanks you for the improved figures and conclusions section. The paper is ready to be published. A few small notes to improve when uploading final version

Section 5: please use heading conclusion (not summary)

L729 (track-changes version). weakened -> replace with 'decreased'

L730: lightly -> replace with 'slightly'

Fig 3: please use white text in panel (b)

Congratulations and thanks for publishing your work in Hess

Best

Thom Bogaard

**Response:** Thank you for your letter and for the decision ( Publish subject to technical corrections) concerning our manuscript (ID: hess-2020-412).

We are so sorry for the inappropriate revision. We have revised the two words in L729, 730. also, we use white text in panel (b), and "Summary" was replaced with "6 Conclusions". Thank you very much. After several revisions according to the valuable comments from editor and reviewers, our manuscript has been improved obviously. Thanks a lot!